

# *tobac* v1.5: Introducing Fast 3D Tracking, Splits and Mergers, and Other Enhancements for Identifying and Analysing Meteorological Phenomena

G. Alexander Sokolowsky[1*], Sean W. Freeman[1*&], William K. Jones[2], Julia Kukulies[3], Fabian Senf[4], Peter J. Marinescu[1,5], Max Heikenfeld[2], Kelcy N. Brunner[6], Eric C. Bruning[6], Scott M. Collis[7], Robert C. Jackson[7], Gabrielle R. Leung[1], Nils Pfeifer[4], Bhupendra A. Raut[7], Stephen M. Saleeby[1], Philip Stier[2], Susan C. van den Heever[1]

[1]Department of Atmospheric Science, Colorado State University, Fort Collins, CO, 80523, USA
[2]Atmospheric, Oceanic & Planetary Physics, Department of Physics, University of Oxford, Oxford, UK
[3]Regional Climate Group, Department of Earth Sciences, University of Gothenburg, Gothenburg, Sweden
[4]Leibniz Institute of Tropospheric Research, Leipzig, Germany
[5]Cooperative Institute for Research in the Atmosphere, Colorado State University, Fort Collins, CO, USA
[6]Department of Geosciences, Atmospheric Science Group, Texas Tech University, Lubbock, TX, USA
[7]Environmental Sciences Division, Argonne National Laboratory, Argonne, IL, USA

\* These authors contributed equally to this work
&Now at: Department of Atmospheric and Earth Science, The University of Alabama in Huntsville, Huntsville, AL 35899, USA

*Correspondence to*: G. Alexander Sokolowsky (g.alex.sokolowsky@colostate.edu)

**Abstract.** There is a continuously increasing need for reliable feature detection and tracking tools based on objective analysis principles for use with meteorological data. Many tools have been developed over the previous two decades that attempt to address this need, but most have limitations on the type of data they can be used with; computational and/or memory expenses that make them unwieldy with larger datasets; or require some form of data reduction prior to use that limits the tool's utility. The Tracking and Object-Based Analysis of Clouds (*tobac*) Python package is a modular, open-source tool that improves on the overall generality and utility of past tools. A number of scientific improvements (three spatial dimensions, splits and mergers of features, an internal spectral filtering tool) and procedural enhancements (increased computational efficiency, internal regridding of data, and treatments for periodic boundary conditions) have been included in *tobac* as a part of the *tobac* v1.5 update. These improvements have made *tobac* one of the most robust, powerful, and flexible identification and tracking tools in our field to date and expand its potential use in other fields. Future plans for *tobac* v2 are also discussed.

## 1 Introduction

There has been a great deal of recent interest in robust, large-scale objective identification and tracking of clouds and other meteorological features (e.g., Heus and Seifert, 2013; Hu et al., 2019; Núñez Ocasio et al., 2020). As atmospheric phenomena of interest are nearly always in motion due to diffusive, advective, dynamic, and thermodynamic processes, there is substantial



utility in tracking frameworks for atmospheric data in general. A moving frame of reference allows one to look at the phenomena from a Lagrangian perspective. Clouds are one such phenomenon for which tracking is useful. Clouds are near-

ubiquitous features in the Earth's atmosphere and play critical roles not only in tropospheric heat and moisture transport (e.g. Malkus, 1958), but also with respect to scattering of solar radiation and absorption/emission of infrared radiation in the context of the global climate (Stephens and L'Ecuyer, 2015). Convective clouds and cloud systems can range in size from tens of meters to hundreds of km; can exist for as short as a few minutes and as long as several days; exhibit a wide variety of morphological characteristics; and undergo complex lifecycles that have a growing initiation stage, a quasi-steady-state mature

stage, and a collapsing decay stage (Cotton et al., 2011). All of these elements make clouds prime candidates for objective analysis techniques, which has been successfully demonstrated in recent cloud tracking studies (e.g. Leung and van den Heever, 2022; Freeman et al., 2023). However, clouds are far from the only meteorological phenomena where robust tracking tools are useful. Convective cold pools, which are density currents that manifest via the evaporation of convective precipitation, can be identified and tracked using atmospheric thermodynamic and dynamic quantities such as temperature or temperature proxies

(e.g. potential temperature), water vapor concentrations, and near-surface wind fields (e.g. Tompkins, 2001; Feng et al., 2015; Drager and van den Heever, 2017; Marinescu et al., 2017; Drager et al., 2020). Atmospheric radiative quantities (e.g., outgoing longwave radiation (OLR)) have clear uses in cloud objective identification (e.g. Gill and Rasmusson, 1983; Weickmann, 1983; Rempel et al., 2017; Senf et al., 2018), but can also be leveraged to detect and track processes such as sea ice evolution (e.g. Singarayer et al. 2006). Lightning mapping systems (Rison et al. 1999, Nag et al. 2015, Bruning et al. 2019) detect the

extent of discharges within thunderstorm cells, and the accumulated extent is a trackable proxy for electrified storm volume. If such tracking tools are made general enough, researchers working outside the realm of atmospheric science may also benefit from them, such as ornithologists or entomologists interested in bird and bug seasonal migration, respectively (e.g. Crewe et al. 2020; Knight et al. 2019). At present, however, only one such tool can address this myriad of uses just described while also being openly developed and extensible by any user: the Tracking and Object-Based Analysis of Clouds (*tobac*; Heikenfeld et

al. 2019), a Python package based in objective analysis principles that uses modern analysis techniques to identify, discretize, and track objects and fields of interest.

The most powerful and unique feature of *tobac* is its ability to use virtually any gridded input dataset and variable – meteorological or not – as input variables, a property we refer to as agnosticity. For example, while *tobac* was initially developed for use with clouds and associated meteorological data (Heikenfeld et al., 2019), with uses including tracking warm-

season deep convective systems and Mesoscale Convective Systems (MCSs) via satellite-observed infrared brightness temperature (e.g. Li et al. 2021; Kukulies et al. 2021), *tobac*'s variable- and grid-agnostic nature has facilitated its use in completely different applications. For example, tracking on quantities such as aerosol concentration (e.g. Bukowski and van den Heever, 2021) and trace gas concentrations and masses (e.g. Zhang et al., 2022) is of enormous use to atmospheric chemists, climate scientists, and others studying movement of such quantities within the atmosphere. *tobac* both draws from



and expands upon the procedures developed in earlier cloud identification and tracking tools, and we have detailed some of the history of tracking tools in the atmospheric sciences below.

Tracking has historically required a great deal of human input and attention due to a lack of computationally efficient methods for the location, assessment, and connection of different features in time. Initial efforts to track clouds from observations were performed by hand (Fujita, 1969), and the need to automate such methods was immediately realised

(Menzel, 2001). One such early method, the Thunderstorm Identification, Tracking, Analysis, and Nowcasting tool (TITAN; Dixon and Weiner, 1993), is a well-designed and powerful approach for the detection and tracking of thunderstorms. While it does incorporate computational analysis of data, it is heavily reliant on physical principles (i.e., it requires specific datasets/variables and can only be used to track certain phenomena) and requires manual assessment of the output due to the computational limitations at the time. As discussed in Dawe and Austin (2012), earlier studies involving tracking of clouds

(e.g. Zhao and Austin, 2005a,b; Heus et al., 2009) required scientists to contribute a great degree of manual/visual selection to the clouds they considered in their studies. This is not only time-consuming to an extent that is impossible to scale for large datasets, but also introduces subjectivity to an analysis that should ideally be objective. Some later publications (e.g. Plant, 2009; Dawe and Austin, 2012; Heus and Seifert, 2013) have more general criteria allowing for automated selection, but exhibit computational or scientific limitations due to their design. Dawe and Austin (2012) tracked clouds as a combination of 3D

liquid water content and buoyancy in 3D space but required computationally expensive determinations of 4D spatiotemporal connectivity and had specific definitions for different cloud components, limiting use to a variety of different cloud types. Heus and Seifert (2013) simultaneously expanded on and improved the tractability of the approach of Dawe and Austin by connecting thermals, cloud envelopes, and precipitation shafts, but reduced the amount of memory needed by projecting these fields into two spatial dimensions and using the vertical dimension as a contiguity check between feature columns. However,

both Dawe and Austin's (2012) and Heus and Seifert's (2013) methods were designed to be used in large eddy simulation (LES) output fields of shallow cumulus with a vertical extent of less than 4 km, thereby limiting the applicability of these methods with cloud systems that exhibit more vertical structure (e.g., layered clouds, deep convection, or slantwise convection) and other datasets that have similarly complex 3D morphology. Gropp and Davenport (2021) recently developed a powerful tracking tool for supercell thunderstorms that was effectively demonstrated at a 3-hourly time resolution (coarser than the

requirements of many tracking tools) but is limited by its focus on supercells and cannot be easily generalized to other cloud types or features due to its inherent design. Similar utility limitations can be seen in the many tracking tools which have incorporated procedures for splits and mergers of tracked objects (e.g. Dixon and Weiner, 1993; Gambheer and Bhat, 2000; Hu et al., 2019; Núñez Ocasio et al., 2020): most of these tools leverage the specific phenomena being detected and tracked in order to construct a definition for the determination of splits and mergers, which preclude such treatments from being used

outside the framework of these particular cases. The Warning Decision Support System–Integrated Information (WDSS-II) data synthesis platform (Lakshmanan et al., 2007) includes a multi-scale cell tracking algorithm and cross-sensor fusion capability (Lakshmanan et al., 2009; Lakshmanan and Smith, 2009) that has been widely used for real-time applications in the





US National Weather Service, but is subject to licensing restrictions for that purpose, and is generally closed-source. Some other tools, such as the TempestExtremes package developed by Ullrich and Zarzycki (2017) and the PyFLEXTRKR package developed by Feng et al. (2022), utilize a more general variable and grid framework but lack comprehensive area and volume analysis tools for further investigation of feature-associated data. It is therefore evident that there already is a rich history of different detection, analysis and tracking tools in the atmospheric sciences, and as such, *tobac* v1.2 strives to utilize as many of the strengths of these pre-existing tools as possible while broadening science applications and optimizing procedures to result in a more general and powerful analysis tool. Additionally, *tobac* was designed to be open-source and modular, and was also developed with open-science principles in mind. These characteristics make it especially unique in conjunction with its variable- and grid-agnosticity. Users can freely download the *tobac* package and modify it as extensively as they please and also have the ability to only use different components of it with other Python packages. The continuous development of the *tobac* package and its detailed, user-friendly documentation have made it increasingly accessible and attractive to atmospheric scientists performing data analysis.

Despite the utility, modularity, and flexibility of *tobac* v1.2, with the increasing resolution of models, identification of new use cases (such as in LES modelling), and the advent of new spaceborne missions such as the National Aeronautics and Space Administration's Atmospheric Observing System (AOS) and Investigation of Convective Updrafts (INCUS) programs and the European Space Agency's EarthCARE program, it became clear that the code base needed to be enhanced from both a scientific and procedural point of view. In order to update *tobac* for these needs, its scientific capabilities were enhanced through the inclusion of the third spatial (vertical) dimension in feature detection and tracking, the processing of feature splits and mergers through time, and tools allowing for spectral smoothing of input data. Additionally, we incorporated more procedural improvements such as increases in computational efficiency, ingestion of multiple data sources on different grids (e.g., performing feature detection on one grid and segmentation on a separate grid), and treatments for periodic boundary conditions (PBCs).

Our goal in this publication is to present each new improvement that has been released as part of *tobac* v1.5. In Section 2, we discuss the strengths and weaknesses of the modular and open source *tobac* v1.2 package with demonstrations of its capabilities, while Section 3 details the scientific improvements. Section 4 presents the procedural enhancements, and Section 5 provides a summary of our changes to *tobac*, concluding thoughts on *tobac* v1.5, and some planned changes which will be included in future releases.

## 2 Overview of *tobac* v1.2

Before elaborating on the new capabilities which have been included in *tobac* v1.5, we begin with a general overview of the design and capabilities of the original *tobac* package, denoted v1.2. *tobac* was first developed through a multi-institutional collaboration (Heikenfeld et al., 2019) in order to provide a modular code base for "tracking and analysing individual clouds in different types of datasets". This package consisted of three primary components: *feature detection*, or the



objective identification of features from minima or maxima in gridded datasets; *segmentation*, or the discretization of the same or different gridded data based on previously detected features; and *tracking,* or the linking of detected features to one another through time. Segmentation and tracking operate independently of each other, but both require feature detection to have been performed on a data field of interest. Hereafter, we will use the term 'feature' to denote phenomena identified using the feature detection module; 'segmented features' or 'segmentation fields' to mean the instantaneous and spatially extensive regions

associated with features by the segmentation module; and 'cell' or 'track' to refer to the line segments produced by spatiotemporally linking features. Do note that the use of 'cell' here does not necessarily mean the kinds of convective cells that comprise thunderstorms, though it can if updrafts are the features of interest.

The procedures contained within *tobac* v1.2 could be performed on any gridded data field of interest, though only segmentation could be performed on data in three spatial dimensions, whereas feature detection and tracking could only be

performed on data in two spatial dimensions, requiring some form of data dimensionality reduction when analysing three-dimensional data. These key elements, demonstrated using a field of radar reflectivity data, can be seen in Figure 1. The details of how these components were constructed are described in Heikenfeld et al. (2019), but we discuss the generalities and how *tobac* can be applied to different use cases within this section.

Feature detection in *tobac* is performed by first establishing one or more contiguous regions of gridded data meeting

or exceeding a threshold, as well as satisfying additional criteria such as a user-set minimum size. These regions are then saved as unique single-point identifiers. The point location associated with each identifier can be set by users to either be geometric centroids, weighted-difference positions, or extrema within the data. Should the user provide multiple thresholds, features detected at a higher-magnitude threshold that exist within a lower-threshold region of features supersede and replace the feature(s) detected at the lower threshold (e.g., Heikenfeld et al., 2019, their Figure 2). This multi-threshold capability allows

for the identification of greater-magnitude data existing within a lower-magnitude data region without losing the sensitivity to lower-magnitude data. For example, using multiple thresholds on a modelled vertical velocity field enables the detection of deep convective updrafts within a broader, weaker updraft region as well as isolated, weak boundary layer thermals. An illustration of feature detection being performed on gridded NEXRAD radar reflectivity data obtained during the CSU Convective Cloud Outflows and UpDrafts Experiment (C³LOUD-Ex; van den Heever et al. 2021) can be seen in Figure 1a-b.

In this figure, convective storms in a grouping near Cheyenne, WY (Fig. 1a) are identified using a radar reflectivity threshold of 30 dBZ with the weighted-difference method. Each of these storms is labelled as a single-point feature, marked in Fig. 1b. Once features have been identified, the additional components of *tobac* – i.e., segmentation and tracking – can be utilized.

The segmentation approach within *tobac* v1.2 begins with a previously identified set of *tobac* features. Where the feature detection procedure reduces contiguous regions of data to single points, segmentation discretizes a full volume or

surface area associated with each of these identified features. For both 2D and 3D segmentation, the skimage.segmentation.watershed procedure (Carpenter et al., 2006; van der Walt et al., 2014) is used. In this method, feature locations are used to place 'seeds' in the data, which are expanded outwards progressively down the gradient of the data in the



same manner that fluid would flow – hence the term 'watershedding' (see e.g., Senf et al., 2018). This allows for the discretization of data regions pertaining to each feature, even when multiple features exist within the same contiguous data
region. In 2D watershedding, this procedure simply operates in two dimensions, but for 3D watershedding, the entire vertical column where the 2D feature is located has markers placed in it, except where data points do not exceed the segmentation data threshold. When data fields are layered, staggered, discontinuous in height, or otherwise irregular through the vertical dimension, this may lead to some data fields being erroneously segmented together. Such misrepresentations have been identified through quality control of *tobac* v1.2 output and triggered development of improvements. The discretized field, or
"segmentation mask", for each timestep is saved as an array with the same dimensions as the input field. Segmentation fields produced using the 2D radar reflectivity data from our previously selected 2D radar reflectivity features (Figure 1b) are shown in Figure 1c. Each segmented region illustrates a wider and weaker reflectivity field located outside of a greater reflectivity region. These segmented regions are associated with the detected convective cores (features), and most likely indicate rainfall from the larger clouds being driven by the convective cores.

Finally, the tracking procedure within *tobac* v1.2 also requires a previously existing set of *tobac* features. These features are then used with the Python Trackpy package (Allan et al., 2021) to predictively link connected features in time through the Crocker-Grier algorithm (Crocker and Grier, 1996). The presence of this tool within the *tobac* package introduces time evolution to the identified features and also links features to each other. Not only does this allow for the examination of cells throughout their lifetime, but also permits the scrutinization of individual features and of any or all features comprising
the cell, which is highly useful for studying storms, clouds, and other temporally evolving meteorological phenomena. This use of Trackpy and other Python packages demonstrates the modular nature of *tobac*, and its ability to capitalize on software development advances occurring in different communities. This not only enhances the performance of *tobac* itself, but also provides it with the flexibility to be used with Python packages other than those used to develop *tobac*.

Despite the utility and power contained within this tool, *tobac* v1.2 had several important limitations from both a
scientific and procedural standpoint, as touched on in the introduction. The limitation of feature detection and tracking to 2D, as well as the column-based approach to 3D segmentation using 2D features, meant that data fields which did not reduce cleanly into two dimensions (e.g., environments with strong vertical wind shear or layered clouds; deep convective clouds with multiple discontinuous vertical regions producing condensate; tilted convective storms; and intrusions of aerosol layers composed of different species at different altitudes) might have produced untrustworthy or confusing results when analysed
using *tobac* v1.2. The *tobac* v1.2 tracking approach also lacked the ability to identify and process the splits and mergers of features over time, which is an issue that previous researchers developing tracking tools encountered and attempted to address (e.g., Dixon and Weiner, 1993; Hu et al., 2019). Additionally, included data processing tools were limited, with no bandpass or spectral filter techniques included in the *tobac* v1.2 package to smooth or isolate data in noisy fields. From a computational perspective, the original implementation was also not well optimized, with one example of tracking several hundred thousand
features (representing about two weeks of model data at 5 minutes) taking over two weeks to process on a modern server and



requiring substantial increases in computational efficiency to enable tractable usage with large datasets. Using detected features to segment data that exists on a different grid was also more challenging with this version of *tobac*, as it required users to remap these data to a common grid. Finally, *tobac* v1.2 also lacked the ability to compute features, segmentation fields, and tracks on data with PBCs, a common characteristic in idealized numerical models. All of these needs motivated the
improvements that are discussed in the following two sections.

## 3 *tobac* v1.5 – Scientific Improvements

### 3.1 3D feature detection, segmentation and tracking

One of the scientifically consequential improvements to *tobac* made as a part of v1.5 is the addition of the vertical dimension to feature detection and tracking, as well as an overhaul of 3D segmentation. When 3D data are input, contiguity
and spacing of regions within these data are now assessed in all three spatial dimensions versus just the horizontal dimensions in v1.2. Further, the code also supports both uniform and non-uniform vertical grid spacing, allowing for use with modelling and observational data exhibiting either of these common grid structures. Data fields with a 3D input now output additional information on the vertical centre of the feature, using the same centre-finding methods that apply to 2D input. Including these additional data can be used for analyses that depend on vertical information, e.g., defining the vertical structure of updrafts
and downdrafts within convective clouds; identifying intrusions of concentrated aerosol layers; and highlighting vertical layers of elevated environmental stability, to name a few.

In addition to the wider variety of scientific analyses that vertical information enables, these code changes also lead to substantial differences in feature detection output between 3D data and their counterparts reduced to 2D, such as that seen in Figure 2. Here, a model vertical velocity field is used for feature detection of updrafts at 1, 3, 5, and 10 m/s thresholds, with
the 2D reduction being a plan view of the column maximum value. Figure 2a illustrates how much of the vertical structure of a 10 m/s feature in the data (white dots within the coloured isosurfaces) is captured by our new method. Comparison of Figure 2a and 2b shows that 3D features' horizontal positions may differ from their 2D-projected counterparts when the vertical dimension is included in feature detection and positioning. For convective systems with a high degree of 3D organization, such as Quasi-Linear Convective Systems, capturing the third dimension can be important to correctly analyse the microphysical-
thermodynamical-dynamical coupling which governs their evolution.

While 2D feature detection is less computationally expensive than 3D and may be a faster solution that produces comparable results, users may also find that 2D projections of 3D data can lead to erroneous results, such as that illustrated in Figure 3. Here, a cumulus cloud and cirrus cloud existing within a sheared environment are traveling in opposite horizontal directions, with the cumulus cloud also moving upwards in time. Fig. 3a-c depict how *tobac* v1.2 is able to identify the clouds
in the initial scene but fails to track the cumulus cloud due to the cirrus cloud hiding it from view in Fig. 3e due to the two-dimensional framework. This leads to the cirrus cloud being correctly tracked through time, while tracking of the cumulus





cloud is non-existent: its height evolution is missed, and the failure to detect it as a feature in Fig. 3b leads to it being considered as a separate, completely new tracked feature in Fig. 3c. Conversely, Fig. 3d-f depict the time evolution of this scene when 3D motion and detection are considered by *tobac* v1.5: not only are these two discrete clouds recognized, identified, and tracked

correctly in time, but the vertical displacement of the cumulus cloud is also apparent in its track. Thus, a possible error arising from collapsing 3D data to 2D is the disappearance of 3D features.

Unlike with feature detection, the segmentation routine in *tobac* v1.2 already has some capabilities for 3D data processing, as discussed in the previous section. The column-based 3D segmentation approach used in v1.2 – where the entire vertical column at a feature location is seeded with markers for watershedding (the segmented regions are identified growing

outward from the seeds) - works reasonably well for 2D features when the 3D field being segmented does not exhibit much vertical stratification or tilting. However, seeding the full column is not a rigorous approach when we have the feature's vertical position, as with 3D-detected features. As such, we have introduced a new "box seeding" method which seeds a box of user-defined size in each dimension centred at the 3D location of the feature. This eliminates the possibility of spuriously connected grid points arising from seeding an entire column, and ensures that features which are close in 2D space but exhibit greater

vertical separation do not unduly influence each other's segmentation masks. A depiction of the differences in 3D segmentation from each method can be seen in the schematic pictured in Figure 4. This figure depicts a multi-layered cloud field of cumulus, altostratus, and cumulonimbus, where segmentation is being performed on total condensate. The top row (Fig. 4a-b) illustrates the use of the older column seeding method and its output, with the bottom row (Fig. 4c-d) visualizing the new box seeding method and the ensuing segmentation fields. The segmentation masks produced are markedly different between 4b, d, with

there being clear examples of misattributed segmentation fields. For example, the cumulonimbus cloud (red feature) is broken up into multiple segmented regions arising from the features associated with the altostratus (cyan and magenta features) and cumulus (orange feature) clouds located closer to the surface.

A further example of this procedure using LES model data is seen in Figure 5: Fig. 5a shows the segmentation mask volume produced via column seeding, while Fig. 5b's segmentation mask was produced by box seeding covering 5x5x5 grid

points. Fig. 5a's segmentation mask exhibits anomalous grid points extending up and down from the main volume, including a disconnected region of points about 1 km above the rest of the mask, which are unphysical and do not manifest in the box-seeded mask seen in Fig. 5b. Since minimizing user effort for objective analysis is one of the key motivators for the development of *tobac* and other comparable tools, use of the box seeding approach is a better approach when users have the choice to do so. This benefits the science itself by making analyses more consistent and less subjective and also permits layered

feature detection and segmentation. However, since 3D data are not always available and the box method may not be strictly necessary for every case when it is available, we allow users to choose between the older column seeding method and the new box seeding method.

Finally, the 3D modifications to tracking are more comparable to those seen for feature detection than segmentation but include similarly powerful advances to both of these components. Since tracking in *tobac* is largely processed using



Trackpy functions, we leveraged the pre-existing Trackpy framework to perform 3D tracking, thereby keeping results both internally consistent and enabling the use of the same general methodology, regardless of whether the user is tracking on 2D or 3D data. Further, our implementation of 3D tracking in *tobac* v1.5 allows users to track on data in 3D with irregularly spaced vertical grids (e.g., stretched model grids) without requiring the user to re-grid the data. Figure 6 illustrates the use of 3D tracking on NEXRAD radar reflectivity data. In these data, a convective core that tilts with height is detected and tracked,

showing the movement of feature position in both horizontal space (Fig 6a-c) and vertical space (Fig 6d-f). Since the feature tilts from west to east with height, the actual 3D centroid appears to be misplaced in the 2 km AGL plan view (Fig 6a-c), but the vertical cross section (Fig 6d-f) indicates that our detected feature centroid is indeed located here in 2D projected space due to its centre being at roughly 4 km AGL. Thus, identifying the centres of such features and discretizing associated data fields are much more realistic with 3D feature detection and box seeding, respectively. As tracking brings temporal evolution

into feature analyses, incorporating the vertical dimension further expands these capabilities by allowing users to assess the change in vertical position over time instead of just the horizontal projected position. For cases where the features of interest are known to exhibit vertical movement as part of their evolution – such as the growth and decay of convective clouds, the development of cold pools and hail cores in thunderstorms, and mechanical lofting of aerosols such as dust or pollen – the importance of including this dimension is essential in feature assessments over their life cycles.

**3.2 Cell Splits and Mergers**

Another key scientific improvement made in this version of *tobac* was the introduction of a procedure for the handling of cell splits and mergers. Splits and mergers are common in atmospheric phenomena: convective storms frequently split into distinct cells (e.g. Newton and Katz, 1958; Charba and Sasaki, 1971; Klemp and Wilhelmson, 1978; Bluestein et al., 1990); aggregation of convection has been studied extensively over the past several decades (e.g. Malkus and Scorer, 1955; Masunaga

et al., 2021); aerosol plumes and layers can split into discrete concentrated regions (e.g. Simpson et al., 2003); and even synoptic-scale troughs are understood to merge under specific conditions (e.g. Gaza and Bosart, 1990). These are just a few examples of splits and mergers in the atmosphere and thus there is a clear need for splits and merger processing within *tobac*.

While there is a critical need for representing splits and mergers within *tobac*, the actual implementation of such processing is a complex endeavour that frequently depends on the type of object being tracked. The detection and definition

of split and merger events in meteorological features can be highly sensitive to various factors such as the time interval between observations, the velocity and size of the objects, and their evolution over time. One significant challenge in detecting these events is the sensitivity to the number of objects in the search region and the initial detection criteria, such as thresholding, which can result in "jumping errors" (Lakshmanan and Smith, 2010). In general, for larger objects whose displacements are comparable to their size, the overlapping criteria is considered to be more reliable for detecting merge and split events as it

results in fewer false alarms (Westcott, 1984; Zan et al., 2019; Raut et al., 2021). However, this method can miss events with rapidly evolving systems and longer time intervals between observations (Nuñez Ocasio et. al., 2020). Another popular method





is to predict the object centre and estimate the best path by minimizing the movement of the object centres (Dixon and Wiener, 1993). It is important to consider the trade-offs and limitations of each method depending on the specific application and data being used.

There are a number of existing tools with split/merge capabilities. The TITAN framework of Dixon and Weiner (1993) makes 2D or 3D determinations of splits and mergers of storm tracks using a combination of reflectivity-based detections and comparisons of path lengths between storms identified in one frame versus the next. Gambheer and Bhat (2000) took a simpler approach that utilized the storm centroid positions, storm area, and associated radii to determine tracks, splits and mergers. The tracking algorithm developed by Hu et al. (2019) was designed for use with observed radar echoes and includes very innovative techniques for identification of storm splits and mergers by detecting and tracking maxima in vertically integrated liquid derived from radar volume scans. The Hu et al. technique can be used with a variety of systems: isolated warm-phase convective cells, isolated mixed-phase convective cells, and multicellular convective storms. Núñez Ocasio et al. (2020) developed the Tracking Algorithm for Mesoscale Convective Systems (TAMS), which builds on prior work by utilizing a combination of previously developed techniques such as area-overlapping (which has also been used more recently, e.g. by Feng et al., 2022), Lagrangian centroid projection, and the use of climatological data on Mesoscale Convective System (MCS) propagation speed to account for splits and mergers when tracking these large systems. However, similar to the supercell tracking approach of Gropp and Davenport (2021), most of these tools were designed to be applied to specific phenomena, and are not readily adaptable to other scenarios. The area-overlapping approach, which was also used in the tool developed by Feng et al. (2022), also requires a high temporal resolution and precludes the use of data with time increments too coarse for features to spatiotemporally overlap. Thus, as we introduce the split/merge addition to *tobac* (which is compatible with the 3D improvements presented in Section 3.1), we will discuss both the split/merge algorithm procedure and the object-/storm-specific considerations in the context of the algorithm's tuneable parameters.

       The splitting and merging procedure included in *tobac* v1.5 behaves as an independent post-processing step within the tracking module that users can execute after the initial linking of features into time-continuous cells. Recall that cells are defined here as features that are linked together across continuous timesteps, and thus cannot be identified from just a single timestep of features. As input, this procedure requires the both the individual features and the cells present at a single time. Connectivity trees are used to established which cells are candidates for mergers or splits with one another. First, parent branches, which in this case serve as tracks, are constructed from the different cells, with each cell and feature being associated to a single parent track. This association is performed using a minimum Euclidean distance spanning tree (MEDST), which is a method of connecting pairs of points that minimizes the distance in Euclidean space along a tree connecting these points. These sets of points are then connected using Kruskal's algorithm (Kruskal, 1956), which is implemented here via the open-source Python package NetworkX (Hagberg et al., 2008). We demonstrate this via a generalized depiction of Kruskal's Algorithm in Figure 7. Here, we start with a web of points (Fig. 7a), from which we progressively identify the shortest distances between two unconnected vertices that do not form a loop (Fig. 7b-i). Once all such segments have been accounted for, the





remaining tree of points (Fig. 7j) is our MEDST. In the context of *tobac*, the specific points connected by the algorithm follow a "tail-to-tip" method: the algorithm works by linking the last feature of a cell to the first feature of a nearby cell. We take the first and last feature of all cells, and then find the distance between each last feature and each first feature at a given timestep for all timesteps. This distance is the weight of the branches in the MEDST. Before further processing these paired points (i.e., the location of the last feature in a cell and the first feature in another cell), we eliminate sets of points which are too far apart

in time, too far apart in space, and those that belong to the same cell. Implementing these basic limitations as a part of this procedure ensures that connected features are near in time and space and do not split or merge with themselves.

Pruning the MEDST results in sub-trees that correspond to the parent tracks of each cell. Each parent track includes one or more associated cells, so that the number of cells is always equal to or greater than the number of parent tracks. Each parent track is assigned a unique integer ID, which is recorded as the parent of each cell in the cell output DataFrame. Since

each feature is also associated with a cell, they are implicitly assigned a parent track ID. Further processing of each parent track can be performed to calculate summary properties such as the number of child cells, the total track length across all cell tracks, the track duration between the first and last feature, and other characteristics of interest. With these new capabilities, *tobac* can now be used to analyze metrics such as the aggregation or splitting of cloud systems (e.g., convective aggregation and supercell splitting into left movers and right movers, respectively); the initiation of discrete convective updrafts due to

mechanical or thermodynamic forcing along outflow boundaries; and constructive and destructive interaction of atmospheric waves that it could not quantify without this framework.

Following our explanation of how the procedure works, we demonstrate its conceptual use in Figure 8. Here, three different cells have been identified from a number of features exceeding 15 dBZ. At time $t_2$, the feature in Cell 1 is identified to also be the spatiotemporal progression of the feature of Cell 2 at time $t_2$. Thus, the merging criteria are met and Cells 1 and

2 are found to have merged. In contrast, Cell 3 stays a distance from the other two cells and is not found to have met the merging criteria with other cells at any point.

As stated previously, the distance limit is critically important to consider in concert with storm mode when assessing splitting and merging. Oftentimes, as Quasi-Linear Convective Systems (QLCSs; e.g. Weisman and Davis, 1998) mature, individual cells will merge over a large distance as the line becomes more coherent (Figure 9). The distance parameter must

be carefully selected in cases such as this, for similar reasons as were discussed in the context of TITAN: while linking the merging cells together with a large value of the parameter is possible, too large of a parameter will lead the MEDST to join smaller cells which meet the criteria but do not physically merge. Figure 9 illustrates a QLCS scenario where the distance parameter is too small, as the cells are physically merging into a convective line but the feature centroids are beyond the prescribed distance criteria. Nonetheless, when the proper considerations with the splits/mergers tool are taken, the scientific

analyses it enables greatly broaden the capabilities of *tobac*.



### 3.3 Spectral Filtering Tool

In addition to the scientific benefits of expanding the dimensionality of *tobac* and enabling it to process splits and mergers of tracked objects, the addition of new data processing tools also expands scientific utility. While *tobac* v1.2 already included some methods for smoothing of data, certain observational and model fields may still be too noisy for these preexisting tools to be useful (i.e., environmental noise that hides the presence of contiguous features), making the use of feature detection and other *tobac* procedures more challenging without additional data processing methods. In order to streamline working with such data, a new spectral filtering tool has been incorporated into *tobac* as part of the v1.5 update. This tool is designed to facilitate the identification of phenomena at specific spatial scales (e.g., the MJO, equatorial waves, atmospheric rivers, mesoscale vortices, etc), and to remove small-scale noise in high-resolution data. For example, with the MJO, sub-mesoscale wind fluctuations might obscure the overall propagation of the convectively active envelope.

The spectral filtering tool works by first performing a discrete cosine transform (DCT) on 2D atmospheric fields, representing them in spectral space as a sum of cosine functions with different frequencies (Denis et al. 2008). This approach allows for the robust isolation of specific frequencies which correspond to phenomena of interest from the dataset. The resulting spectral coefficients correspond to normalized wavenumbers that can be converted to actual wavelengths, which are then used in the construction of a bandpass filter that has the same shape as these spectral coefficients in wavelength/wavenumber space. The bandpass filter can be constructed to be low-pass, high-pass, or a different configuration. Multiplying this filter with the spectral coefficients removes wavelengths outside of the user-specified band, which can then be converted back to the original domain via inverse DCT. A visualisation of atmospheric data and the spectral elements used for filtering are demonstrated in Figures 10 and 11.

Fig. 10a displays the initial 2D input field (here, a WRF relative vorticity dataset), Fig. 10b illustrates the transformation of the data in Fig. 10a to spectral space, and Fig. 10c-d show the construction of 1D and 2D bandpass filters for wavelengths between 400 and 1000 km. The results from applying such filtering to an ERA5 vertically integrated water transport dataset and a WRF relative vorticity dataset are shown in Figure 11. The original, pre-filtered fields of ERA5 and WRF data, respectively, are illustrated in Figs. 11a and 11c, while Figs. 11b and 11d illustrate the same corresponding fields after utilization of the filter. It is clear from Figure 11b,d that the application of the spectral filtering uncovers large-scale spatial patterns obscured by fine-scale noise in the original data.

This filtering approach can be leveraged to identify a wide variety of atmospheric phenomena across different spatiotemporal scales and frequencies, such as the many oscillatory phenomena identified in OLR power spectra by Wheeler and Kiladis (1999). Inclusion of the filtering tool in *tobac* v1.5 clearly expands the package's utility while reducing the amount of extra work needed for end users to pre-process data of interest. This technique has previously been used along with *tobac* to identify mesoscale vortices in convective permitting climate simulations (e.g., Kukulies et al. 2022, in review).

Overall, the 3D implementation, the splits/mergers procedure, and the spectral filtering tool comprehensively address many needs of the tracking community (as evidenced by the multitude of tools and capabilities described in the introduction)



and add a great deal of scientific power to *tobac*. These new features expand on the types and dimensionality of contiguous structures that *tobac* can identify within datasets, allowing the tool to be used with more dynamically evolving phenomena, and providing an additional level of filtering to isolate atmospheric phenomena of interest. However, additional improvements of *tobac* have also been achieved with the addition of procedural changes such as code optimization, homogenization of grids for different data, and treatment of PBCs, all of which are possible in part due to *tobac*'s modular nature. These procedural adaptations are discussed in the following section.

## 4. *tobac* v1.5 – Procedural Improvements

### 4.1 Code Optimization

Several inefficiencies were identified across the body of code – for example, a loop in the tracking module would iterate a number of times equal to the square of the number of features, as opposed to just the number of features - and subsequently, alterations were made to each module to enhance their overall computational speed. Making these changes led to speedups on the order of 100x for feature detection and 1,000,000x or more for tracking. The scaling of these modules' speeds as a function of the number of features, a proxy for data size and complexity, between *tobac* v1.2 and v1.5 can be seen in Figure 12, with feature detection in Fig. 12a and tracking in Fig. 12b. To provide a single example of what this means from a practical perspective, performing feature detection on a full day of GOES-16 IR data (1500 by 2500 spatial grid points, 288 time steps) only takes about a minute of computing time with *tobac* v1.5, whereas it originally took around an hour with *tobac* v1.2 using the same computing platform. This has significant implications for the tractability of using *tobac* v1.5 with larger datasets: analyses on especially large datasets (10s-100s of TB) that would take weeks to perform with *tobac* v1.2 now only take hours to days, which expedites the research that can be conducted with this tool.

### 4.2 Remapping Data on Different Grids

Beyond recognizing that the efficiency of *tobac* needed to be improved to make certain analyses tractable from a computational processing point of view, we also understood that researchers working with data from different sources often have a need to combine these datasets in some way. This process can be complicated by observing platform nuances such as viewing angle and field of view; temporal frequency; spatial resolution; and the dynamic range of the data. Issues such as differing fields of view and spatial resolution have particularly strong implications for the uses of objective analysis tools like *tobac* due to the projection of data onto different spatial grids. Within the framework of *tobac*, we have introduced a new function which allows for the combination of datasets (both modelled, both observational, or a mix of the two) so that *tobac* can be more easily used with a combination of different datasets and types.

This new remapping tool allows for the user to identify features and track on one dataset on one grid (e.g., ground-based radar), and then identify the spatial extent of the features via *tobac*'s segmentation routines on a different dataset on a



different grid (e.g., satellite). Instead of regridding the data internally, this tool instead remaps the feature centroids identified

by feature detection onto the new grid, allowing segmentation to proceed as normal at the full resolution of the new grid. To perform this, *tobac* uses the latitude and longitude of each identified feature point, then employs a Ball Tree (using the Scikit Learn package; Pedregosa et al., 2011) to find the closest point in space to the identified feature location on the new grid. Once this is complete, the user can perform segmentation as normal on the new grid.

One case for the use of the remapping tool is in observational analysis of convection via radar and satellite datasets,

which we demonstrate in Figure 13. Features detected from NEXRAD reflectivity data exceeding a 30 dBZ threshold are shown in Figure 13a. These features are then used as markers to segment a GOES-16 satellite-observed brightness temperature dataset, pictured in Fig. 13b. The satellite brightness temperature data have been remapped to the same grid as the radar data (not incorporating parallax effects) prior to performing the segmentation process, so that features are correctly located within the segmentation field of interest. Ultimately, the segmentation outlines shown in Fig. 13b depict the anvils corresponding to

each radar reflectivity feature, except for the top-right feature marked by the grey dot in Fig. 13b, which is a convective core that does not yet have an associated anvil.

## 4.3 PBC (Periodic Boundary Condition) Treatments

Idealized numerical models and LES often utilize PBCs in order to isolate simulations from external forcings and reduce the influence of the lateral model boundaries on the simulation behaviour. With PBCs, phenomena flowing out of one

end of the model boundary simply re-enter the domain at the opposite boundary for that dimension. However, v1.2 of *tobac* did not have any capabilities for recognizing the continuity of features, segmentation masks, or cell tracks which passed through boundaries or were split into multiple parts by boundaries, and the code base required these improvements for use with model configurations including PBCs in one or both lateral dimensions.

Most of the changes needed for PBC treatments in feature detection lie within the identification of contiguous regions

separated by an artificial boundary and the positioning of features which exist across both sides of a boundary. In the original v1.2 procedure, a failure to recognize when contiguous fields are split by artificial model boundaries leads to an erroneous multiplication of detected features at these boundaries, which further cascades into unphysical segmentation fields and cell tracks. A depiction of PBC feature detection with *tobac* v1.2 and *tobac* v1.5 being performed on an LES model 2D column maximum vertical velocity field can be seen in Figure 14. Fig. 14a shows the overall data field (with values less than 0.5 m/s

masked in grey), and Fig 14b visualizes the initial field of labelled regions identified at a 0.5 m/s threshold prior to utilizing our PBC treatment. Fig. 14b contains a total of 6 different regions due to the multiple boundary crossings exhibited by this vertical velocity field and would produce 6 different features (instead of the singular feature that it is) if a PBC treatment was not applied. After performing the new PBC treatment which overwrites the labelled fields, the resulting unified label can be seen in Fig. 14c, which correctly identifies the object as a single feature. Utilizing the PBC treatment in the zonal direction

also facilitates the use of *tobac* with some global model and observational datasets and represents the first steps towards



enabling global tracking. The PBC treatment for segmentation largely follows the same principles as that for feature detection, except it requires adjustments, rather than complete unifications, to be performed when segmentation masks collide at a model boundary. Beyond these, the PBC procedures for feature detection and segmentation are quite similar.

The tracking procedure for PBCs differs from that for both feature detection and segmentation due to the key purpose of the PBC treatment being to link cell tracks that already exist. Provided that one has performed the PBC treatment within feature detection, propagating features will be crossing boundaries in a smooth manner without the introduction of specious features. An example of the PBC tracking approach can be seen in Figure 15: Fig. 15a displays the erroneous recognition of two distinct cell tracks from an evolving feature crossing the periodic boundary, while Fig. 15b shows the correct identification of a single cell track with the PBC tracking approach. This new capability enables a much more robust assessment of cloud lifecycles and other such temporal processes in models with PBCs that would otherwise produce a disjoint or garbled picture with non-PBC tracking. This becomes increasingly important with smaller domains where boundary crossings are more frequent. As discussed above in relation to feature detection, this PBC code is an important step towards the implementation of global feature detection, segmentation, and tracking in *tobac*. At present, cylindrical (zonal) global tracking (which can be used on Global Precipitation Mission data, for example) is enabled within this framework, but features near or crossing over the poles are still an issue that must be addressed in future versions of this package.

## 5. Summary and Conclusions

Our overall goals for the improvements to *tobac* detailed within this manuscript were to enhance the package's scientific capabilities and utility, improve its efficiency, and incorporate new tools for data processing and more complex analyses. The inclusion of these changes in *tobac* v1.5, as well as the previously existing flexibility and modularity of *tobac* v1.2 and its variable- and grid-agnostic (i.e., capable of working on any gridded dataset) nature, make *tobac* simultaneously one of the most powerful and malleable objective analysis tools that presently exist in our field.

From a scientific point of view, the inclusion of the vertical dimension in *tobac* v1.5 allows for identification, discretization, and tracking of more complex and multidimensional meteorological structures, which could not be performed in *tobac* v1.2. It also allows users to better capture the spatiotemporal evolution of clustered phenomena that are difficult to isolate in 2D projections of 3D data. Further, the processing of mergers and splits within *tobac*'s tracking module greatly enhances the ability to assess the lifecycles of cloud systems that exhibit such processes, without requiring additional record-keeping and data processing by the user. The included spectral filtering tool also improves the scientific utility of *tobac* by providing a method for users to isolate specific frequencies of interest in the data they are using, precluding the need for external data processing or the use of datasets that have already been smoothed.

The procedural enhancements made to *tobac* as a part of v1.5 have also led to a vast expansion in the capabilities of this package. First and arguably foremost, the computational efficiency improvements, ranging from 100x to over 1,000,000x increases in processing speed depending on the module being used and the nature of the data being analysed, allow users to



conduct analyses in far less time than was possible before. Such efficiency improvements allow users to leverage higher resolution data and overall larger datasets than *tobac* could reasonably manage previously. The data regridding procedures that
are now included also enable the combined use of multiple different datasets existing on different grids. New applications that this procedure enables include tracking convective cores on radar while simultaneously identifying anvil regions with satellite data; and comparing modelled lofting of dust in haboob events with satellite observations of the overall dust outflow. Finally, adding the capability to recognize and robustly address PBCs has also widened the utility of *tobac* by enabling its use with applicable model data. PBCs are commonly used in idealized and LES models, which would be prime candidates to analyse
using the older *tobac v1.2* if they did not have these boundary conditions.

Although we have made many modifications to the *tobac* code base as a part of v1.5, future updates are already being developed as part of the next major release, *tobac* v2, and an active, international community of developers continue to maintain its code base. Some of the key elements that are planned for the next major release include integration with the TiNT is not TITAN (TiNT; Raut et al., 2021) tracking package and a transition away from *tobac*'s current memory-intensive data
structures to data structures that allow for out-of-memory computation instead (e.g., Dask from Rocklin, 2015; xarray from Hoyer and Hamman, 2017). The overarching vision for *tobac* v2 is, at present, to continue development and enable better support for Big Data use cases, as well as to move towards data structures that support parallelization for more memory-intensive datasets.

**Code and Data Availability**

The source code for the *tobac* v1.5 package is available on GitHub at https://github.com/tobac-project/tobac and on Zenodo at https://zenodo.org/record/8164675 (tobac Community et al., 2023). All example data for the included *tobac* notebooks is either included in the repository or is automatically downloaded from Amazon Web Services.

**Conflicts of Interest**

The contact author has declared that none of the authors have any conflicts of interest.

**Author Contributions**

GAS and SWF: Software, conceptualization, writing (original draft; review and editing). WKJ, JK, KNB: Software, conceptualization, writing (review and editing). FS: Software, conceptualization, project administration, funding acquisition, writing (review and editing). PJM, MH: Software, conceptualization, writing (review and editing). ECB, SMC: Conceptualization, funding acquisition, writing (review and editing). RCJ: Conceptualization, writing (review and editing).
GRL, NP: Software, validation, writing (review and editing). BAR: Software, writing (review and editing). SMS: Validation, writing (review and editing). PS, SCV: Conceptualization, funding acquisition, supervision, writing (review and editing).



**Acknowledgements**

G. A. Sokolowsky, S.W. Freeman, G. R. Leung, and S.C. van den Heever were supported by the US National Aeronautics and Space Administration (NASA), Grant # 80NSSC18K0149. S.C. van den Heever, S.W. Freeman and P.J. Marinescu also acknowledge support from INCUS, a NASA Earth Venture Mission, funded by NASA's Science Mission Directorate and managed through the Earth System Science Pathfinder Program Office under contract number 80LARC22DA011. J. Kukulies acknowledges funding by Formas (2019-01520). F. Senf acknowledges computational support from the Deutsches Klimarechenzentrum (DKRZ) through the compute projects bb1174 and bb1262. Contributions by R. Jackson, S. Collis, and B. Raut were supported by the U.S. Department of Energy, Office of Science, Office of Biological and Environmental Research's Atmospheric Radiation Measurement User Facility and Atmospheric Systems Research program under contract number DE-AC02-06CH11357. E. C. Bruning and K. N. Brunner were supported by DOE (ARM/ASR) DE-SC0021247, NSF AGS-2019939 and NOAA (VORTEX-SE/USA) NA19OAR4590210 and NA21OAR4590151. S. M. Saleeby acknowledges support from the U.S. Department of Energy's TRACER project, Grant # DE-SC0021160. P. Stier and W. Jones acknowledge funding from the European Research Council (ERC) project RECAP under the European Union's Horizon 2020 research and innovation programme with grant agreement no. 724602 and the ESA Cloud CCI+ project with contract number 4000128637/20/I-NB. P. Stier additionally acknowledges support from the FORCeS and NextGEMs project under the European Union's Horizon 2020 research programme with grant agreements 821205 and 101003470, respectively.

**Financial Support**

This research has been supported by the US National Aeronautics and Space Administration (NASA; Grant # 80NSSC18K0149) and the NASA Science Mission Directorate through the Earth System Science Pathfinder Program Office (Contract # 80LARC22DA011); Formas (2019-01520); the Deutsches Klimarechenzentrum (DKRZ; compute projects bb1174 and bb1262); the US Department of Energy (Contract # DE-AC02-06CH11357; Grants # DE-SC0021247 and DE-SC0021160); the National Science Foundation (Grant # AGS-2019939); NOAA (VORTEX-SE/USA; Grants # NA19OAR4590210 and NA21OAR4590151); the European Research Council (ERC) project RECAP (Grant # 724602); the European Space Agency Cloud CCI+ project (Contract # 4000128637/20/I-NB); the FORCeS and NextGEMs project (Grant # 821205); and the European Union's Horizon 2020 Research Programme (Grant # 101003470).

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

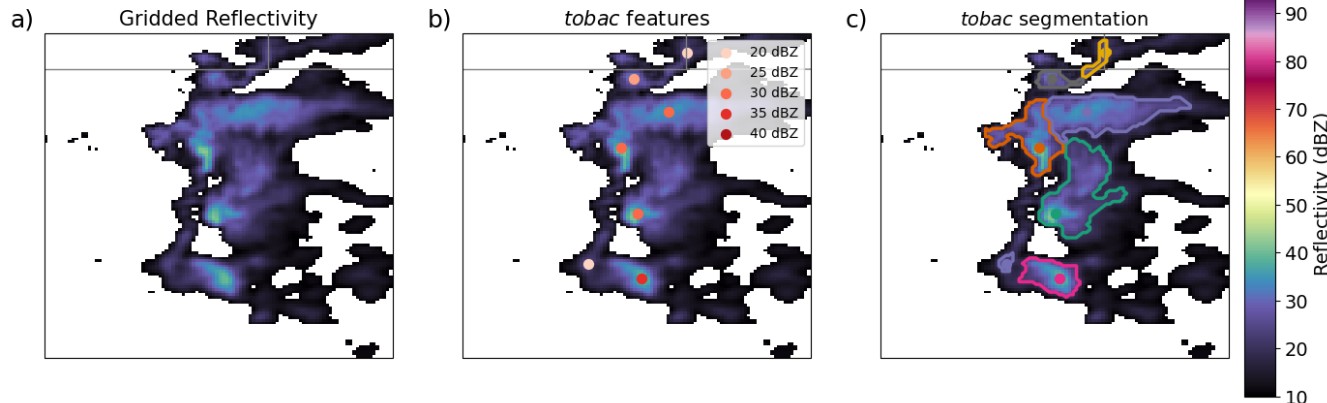

**Figure 1: Demonstration of *tobac* feature detection and segmentation of NEXRAD radar reflectivity data from the Cheyenne, WY radar (located just to the NW of the domain shown here) on 25 May 2017 during the C³LOUD-Ex field campaign (van den Heever 800    et al., 2021). Panel (a) shows the actual radar data, panel (b) displays the objectively identified radar reflectivity features for thresholds of 20, 25, 30, 35, and 40 dBZ as dots with a progressively darker red colour at higher magnitude, and panel (c) shows the reflectivity segmentation regions associated with the features as differently coloured outlines. The straight grey lines depicted in each panel represent state borders.**





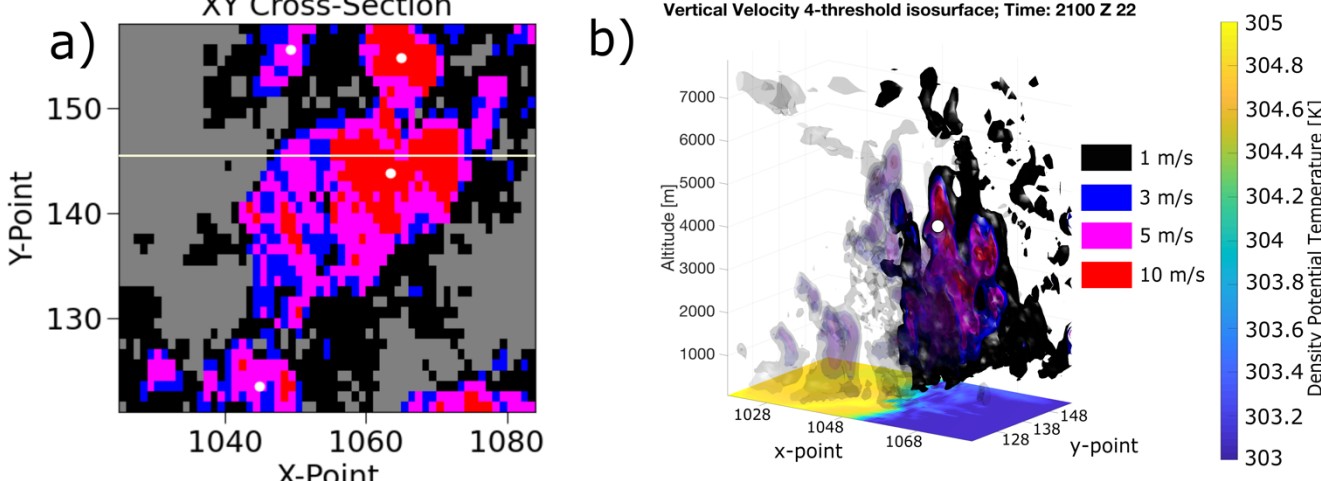

**Figure 2: An illustration comparing cross-sections of 2D and 3D updraft four-threshold feature detection on the same model 3D vertical velocity field. Panel (a) shows the projection of column maximum vertical velocity and the multiple features contained in this area as white dots, while panel (b) shows a cutaway 3D isosurface plot of a 3D updraft detected at the 10 m/s threshold covering the same area as panel (a). Black, blue, magenta, and red shading indicate pixels exceeding the 1 m/s, 3 m/s, 5 m/s, and 10 m/s thresholds, the white dots illustrate feature positions within each cross-section, and the white line in panel (a) represents the location of the front-left cutaway in panel (b), ahead of which (in y-point space) transparent isosurfaces are used to reveal the complex inner structure of the updraft via the opaque isosurfaces. The surface colour shading in panel (b) is surface density potential temperature, and its colours correspond to that seen in the colourbar to the right of panel (b).**





**Figure 3: A depiction of *tobac v1.2* (top row, plan view) and *tobac v1.5* (bottom row, vertical cross section) feature detection and**
**tracking for a scenario with upper-level cirrus moving over a cumulus cloud developing in a sheared environment. Each column's**
**panels are depictions from the same time. The *tobac v1.2* approach pictured in the top row fails to capture the temporal evolution**
**and vertical propagation of the cumulus cloud due to the overlying cirrus, and even incorrectly recognizes the cumulus in panel (c)**
**as a completely new feature and track from its earlier stage in panel (b). In contrast, the *tobac v1.5* approach consistently and**
**continuously identifies each cloud due to their separation in 3D space, resulting in correctly linked cloud tracks for each of the cirrus**
**and cumulus. The coloured circles denote different features at their present times in each panel, with the coloured X's indicating**
**their position at previous times and the dotted lines representing the corresponding tracks. The symbol *t* here denotes a generic**
**starting time, while *Δt* denotes the timestep from scene to scene.**





**Figure 4: A schematic of the new box seeding approach versus the older column seeding approach for tobac 3D segmentation. Panels (a-d) depict a scene comprised of a mix of convective and stratiform clouds, with feature detection and segmentation being performed on a total condensate field. The top two panels (a,b) depict the older column seeding approach, and the bottom two panels (c,d) show the new *tobac* v1.5 box seeding method. The left column shows the positions of the initial features used as segmentation markers as highlighted lines or circles, with the segmentation regions produced from these markers hatched with the corresponding colour in the right column. Note that the midlevel stratiform clouds seeded with cyan and magenta are in front of and behind the cumulonimbus cloud seeded in red, respectively, and would not themselves be seeded or segmented in red with the column seeding method.**





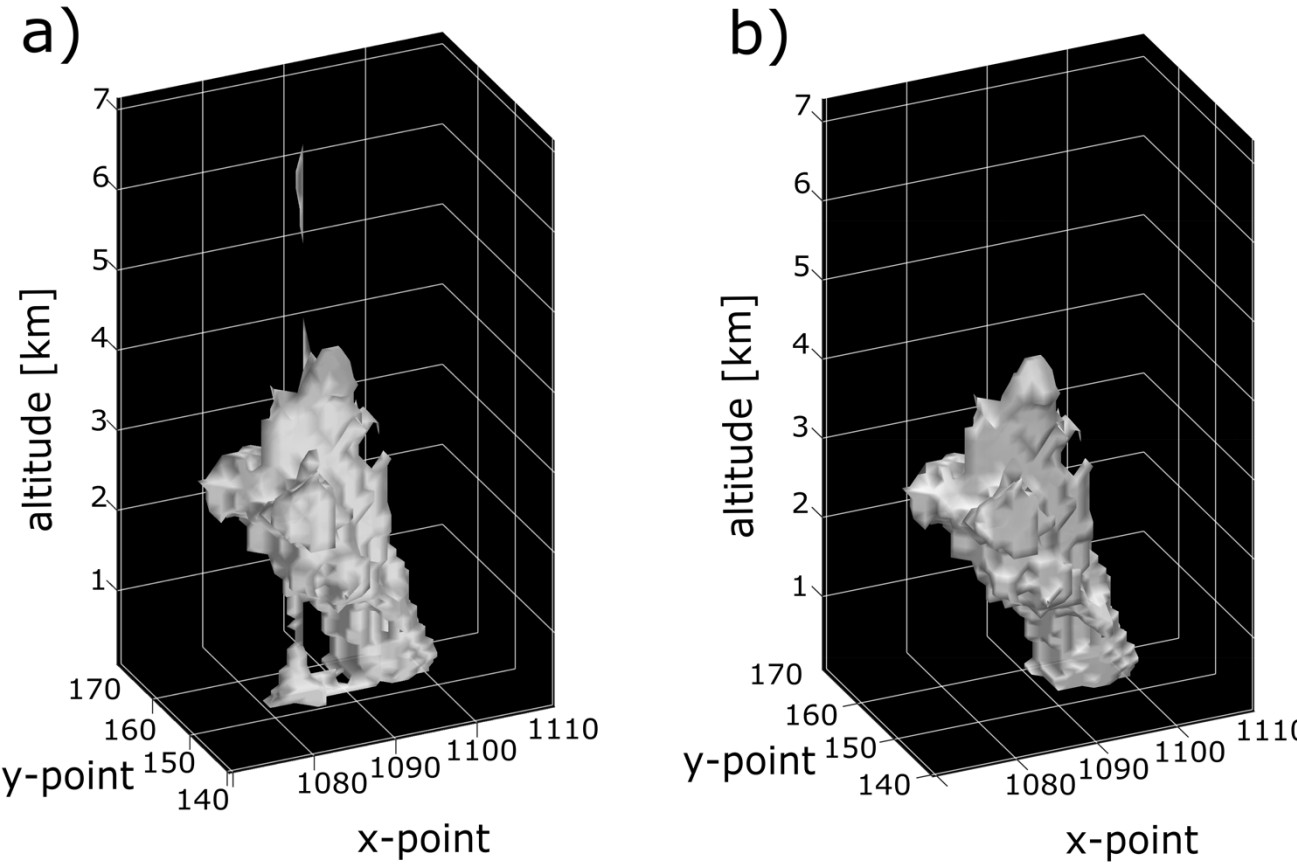

**Figure 5: Demonstration of 3D segmentation using (a) the original "column" versus (b) the "box" seeding method, showing the differences in output produced by the different methods. 3D feature detection was performed on LES numerical model vertical velocity data from the Regional Atmospheric Modeling System (RAMS) v.6.2.14, with segmentation being performed on the corresponding model total condensate field. Segmentation in panel (b) used a uniform box seed size of 5 in x, y, and z.**




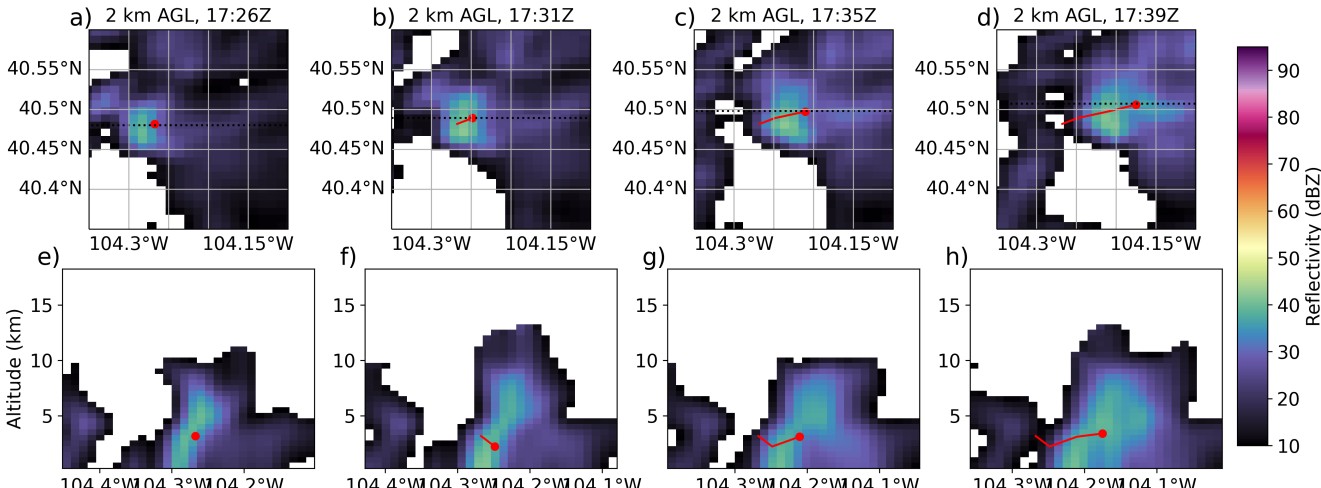

**Figure 6: Demonstration of 3D tracking in** *tobac* **on NEXRAD radar reflectivity data. The top row shows the plan view in latitude-longitude space, while the bottom row consists of latitude-altitude cross sections corresponding to each of the times presented in the plan view above – thus, (a) and (e); (b) and (f); (c) and (g); and (d) and (h) are all pairs. The red dot shows the present feature location, while the red line trailing behind it shows the detected track.**








**Figure 7: A general depiction of Kruskal's Algorithm (Kruskal, 1956) used to construct the minimum Euclidean distance spanning tree (MEDST) for splits and mergers. This illustrates the basic MEDST procedure without consideration for cell start/end points. Panel (a) shows a web of points connected by edges from which we want to identify the MEDST. In each panel from (b) to (i), the**
**shortest edge which has not already been highlighted and does not form a loop between previously connected points is highlighted in red. Panel (j) illustrates the final MEDST produced from this web of points and edges, after pruning the non-highlighted edges.**

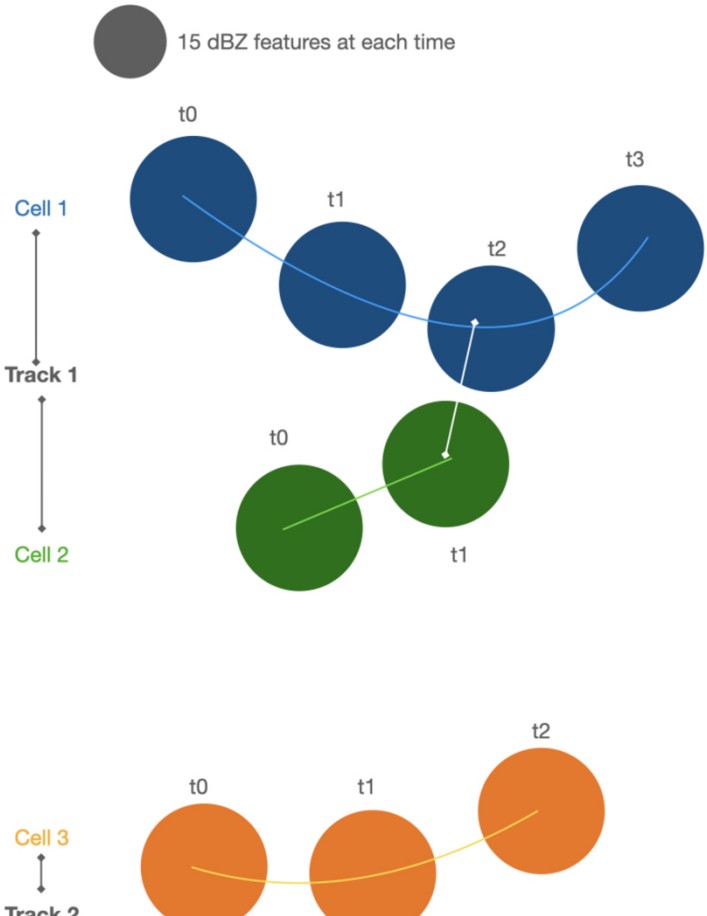

**Figure 8: An illustration of merging cells (Cells 1 and 2) and a standalone cell (Cell 3) as perceived by** *tobac*. **All three cells are**
**comprised of features in radar data which exceeded a 15 dBZ threshold. Cells 1 and 2 can be seen to merge at time** $t_2$, **when merging criteria (size and proximity) are met.**





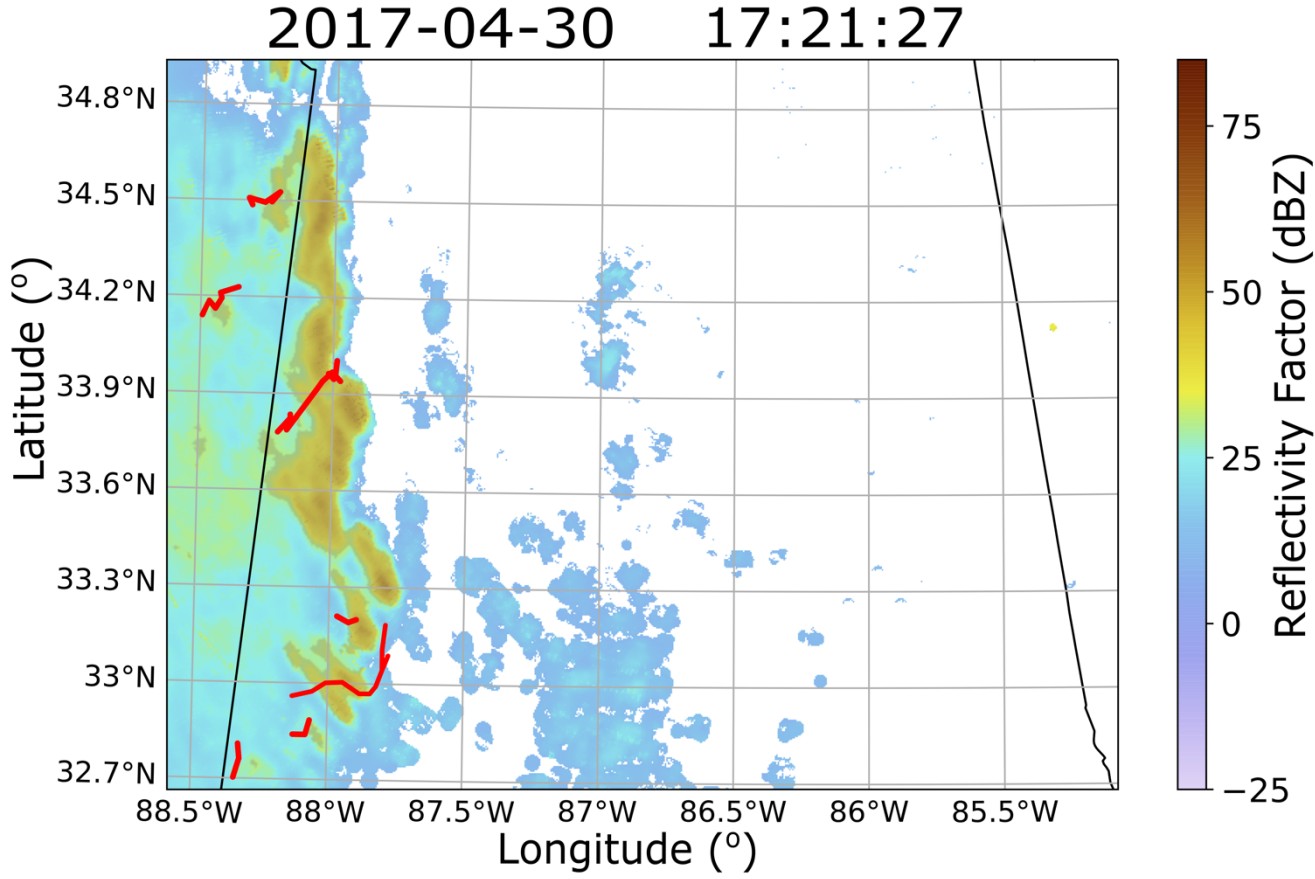

**Figure 9: A Quasi-Linear Convective System (QLCS) in northern Alabama on 30 April 2017, shown on the KHTX NEXRAD radar. The centre track is outside of the distance criteria link with the northern tracks. However, the features, which are detected and tracked at 30 dBZ, encompasses the cells along the line since cells are physically merging with the line.**




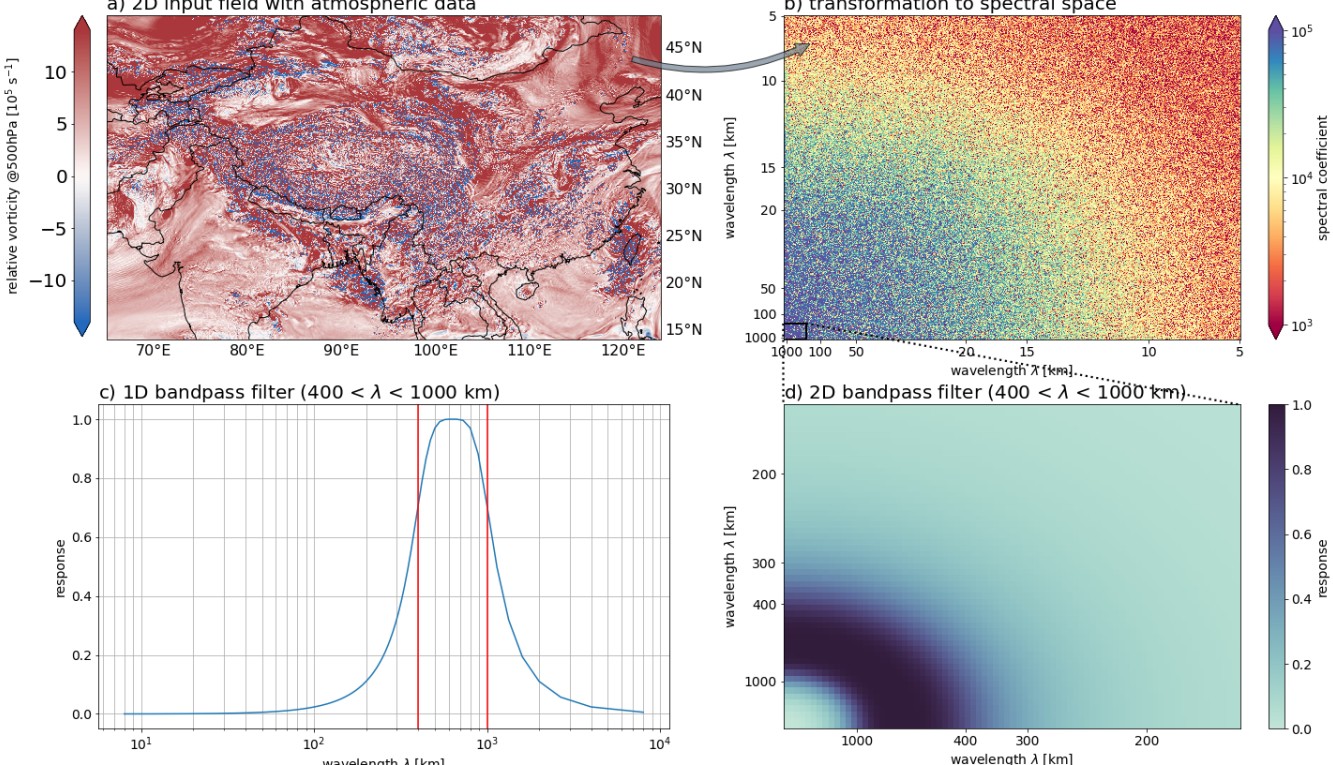

**Figure 10: Visualization of spectral decomposition of atmospheric input fields and construction of a bandpass filter that can be specified by the user and is used to filter the input data. a) 2D input field with atmospheric data at one time step, here: hourly relative vorticity at 500 hPa [$10^5$ s$^{-1}$] of a 4km WRF simulation over South East Asia. b) The same data after the discrete cosine transformation (DCT), represented by spectral coefficients as a function of wavelengths in x and y direction. c) Response of constructed bandpass filter as a function of wavelength. The two red lines indicate the cut-off wavelengths that can be specified by the user (here: 400 and 1000 km). d) Same bandpass filter but in 2D spectral domain with same shape as b) but zoomed in to show the filter response for wavelengths between 400 and 1000 km.**





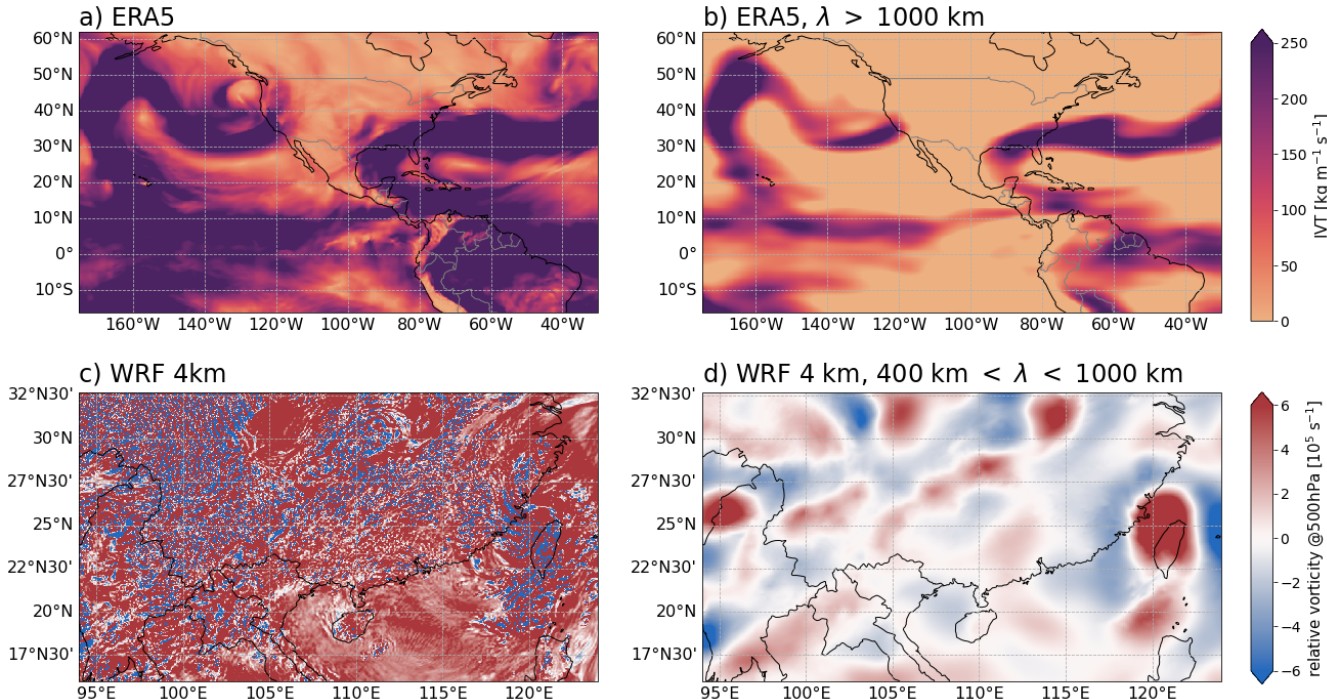

Figure 11: Examples for hourly atmospheric input fields (a, c) and their corresponding spectrally filtered fields (b, d). a) Vertically integrated water vapor transport (IVT) [kg m$^{-1}$ s$^{-1}$] from ERA5 at 2021-01-27 10:00:00 UTC showing an atmospheric river over the San Francisco Bay area  b) Same as in a) but spectrally filtered for wavelengths > 1000 km, c) Relative vorticity at 500 hPa [10$^5$ s$^{-1}$] from a WRF simulation with 4km grid spacing over Southeast Asia for 2008-07-18 05:00:00 UTC (when Typhoon Kalmaegi hit Taiwan) d) Same as in c) but spectrally filtered for wavelengths between 400 and 1000 km. Note that the typhoon over Taiwan only becomes visible in the vorticity field after the filtering has been applied, because the original vorticity field is dominated by sub-mesoscale noise.

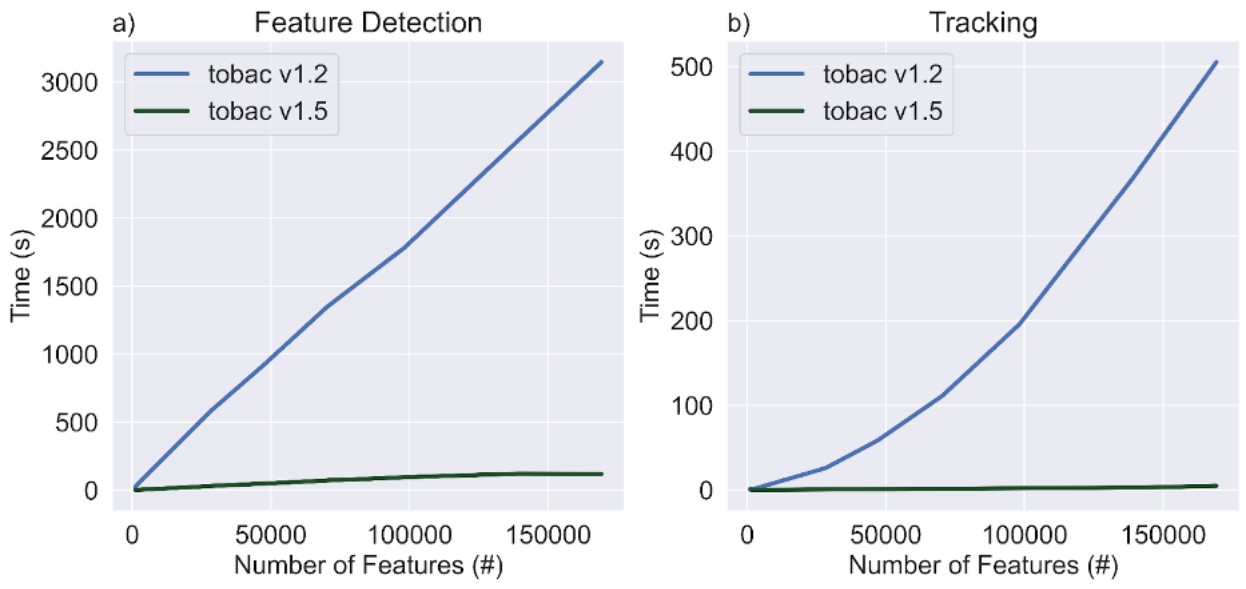



**Figure 12: A benchmark comparison of** *tobac* **speed between version 1.2 (Heikenfeld et al. 2019) and version 1.5, demonstrating the**
**increase in speed using a full day of GOES-16 Channel 10 IR imagery from 12 June 2021 on a) feature detection, with number of features on the abscissa and time taken to run feature detection on the ordinate, and b) as in a, but for tracking.**

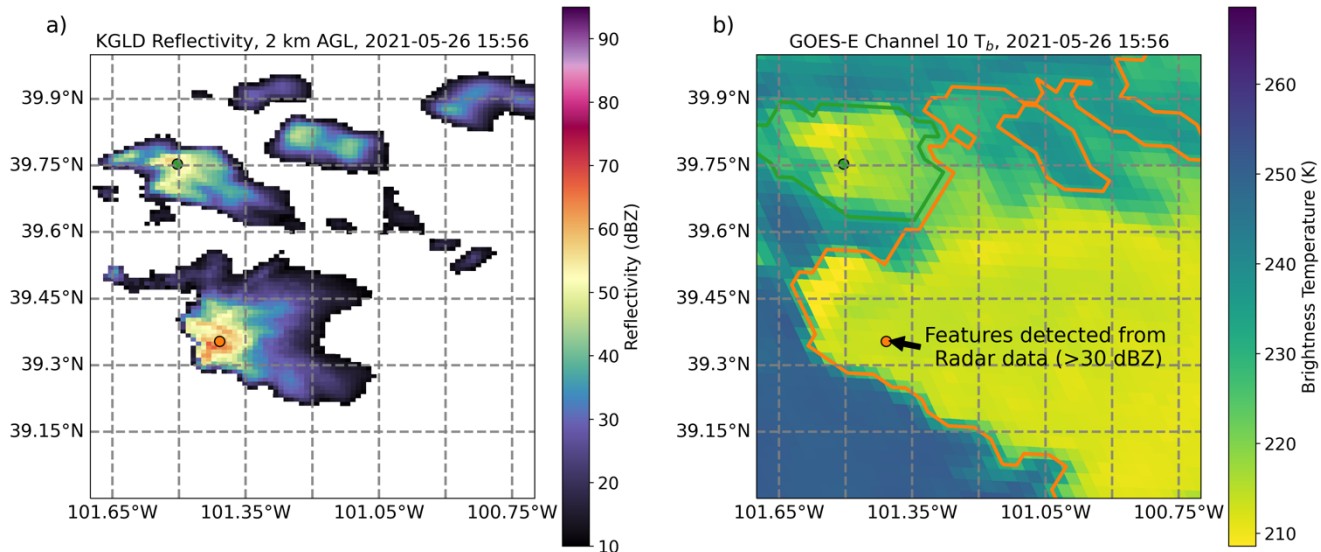

**Figure 13: A depiction of the output from the new procedure for differently gridded data included in** *tobac* **v1.5. Panel (a) shows NEXRAD radar reflectivity in dBZ from the Goodland, KS site at 15:56 UTC on 26 May 2021, as well as the associated features**
**detected at a 30 dBZ threshold marked by grey dots which represent different convective cores. Panel (b) shows GOES-16 satellite observed brightness temperature in K (initially on a different grid from the radar data), as well as the segmentation masks associated with each of these features as differently coloured outlines. The segmentation outlines shown in panel (b) are produced after regridding the satellite data to the same grid as the radar data and depict the upper-level cirrus shields associated with the different convective cores seen in the radar data.**

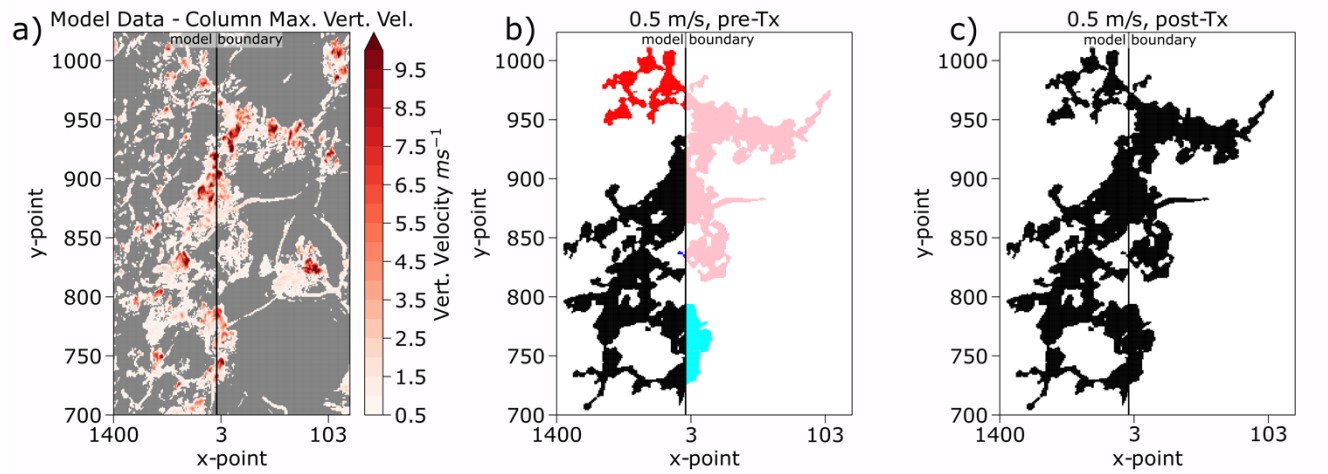


**Figure 14: Illustration of PBC treatment algorithm for feature detection. Panel (a) shows the original column-maximum vertical velocity field (values less than 0.5 m s⁻¹ masked); (b) depicts the six individual feature detection labels produced at a 0.5 m s⁻¹ threshold without the PBC treatment; and (c) presents the correct unified label post-treatment for PBCs.**





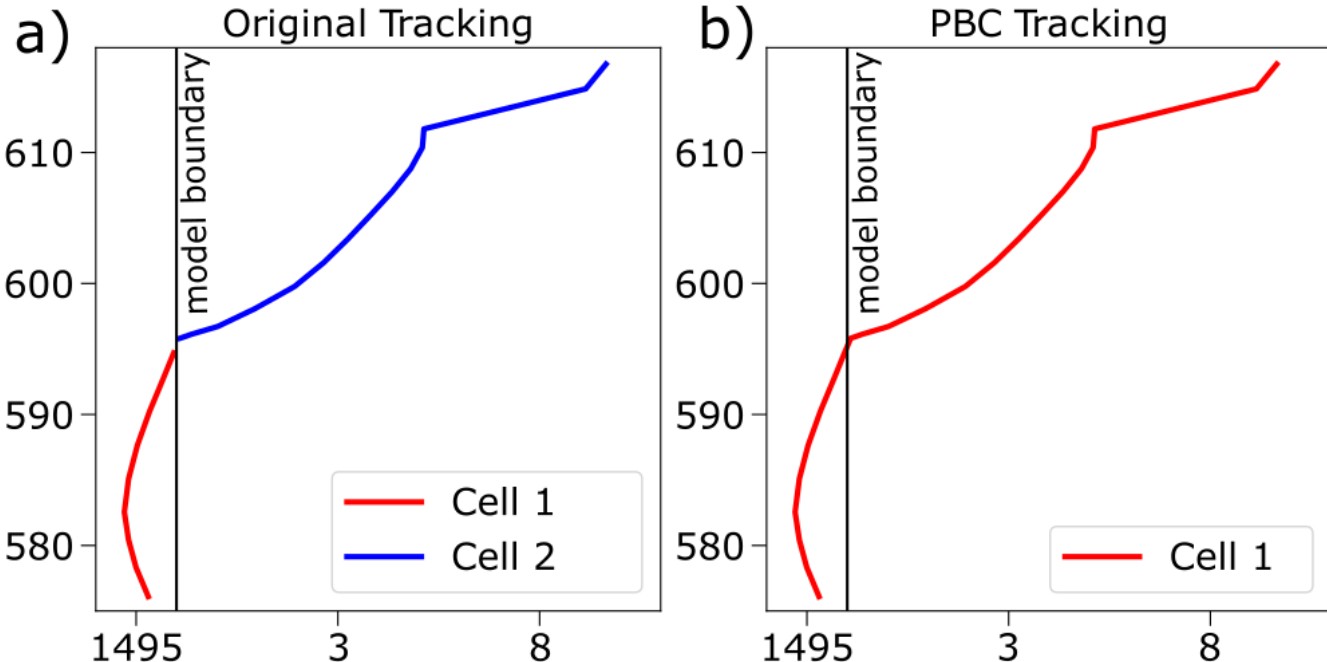

**Figure 15: A depiction of 2D *tobac* tracking with and without accounting for PBCs. Panel (a) shows the two discrete cells that would be identified by *tobac* v1.2 when a feature crosses a boundary; panel (b) illustrates the single, unified cell that is produced with the PBC tracking procedure.**