# Peer review of "tobac v1.5: Introducing Fast 3D Tracking, Splits and Mergers, and Other Enhancements for Identifying and Analysing Meteorological Phenomena"

_EGUsphere, 2023_

## Referee Comment (RC3)

The manuscript introduces the new characteristics of the Tracking and Object-Based Analysis of Clouds (tobac) v1.5 Python package. The updated features include 3-D tracking, splitting and merging, internal spectral filtering, and treatments of periodic boundary conditions. The computational efficiency of tobac v1.5 is also enhanced significantly compared to earlier versions after coding optimization. Overall, the manuscript is well-written, well-organized, and easy to read. Most of the figures are helpful and clear. The topic is within the scope of GMD. The new characteristics of tobac v1.5 are essential to the scientific community and deserve to be introduced in a new manuscript. Here, I have some comments for the authors to improve the manuscript before publication further.

Line 1-3: Could you please use a better title?

Line 32: Delete "diffusive, advective"? Do you think diffusion and advection are neither dynamic nor thermodynamic processes?

Lines 281-282: I don't understand the logic of this sentence. If there are just a few examples of splits and mergers in the atmosphere, why is there a clear need for splits and mergers processing within tobac?

Line 292: What do you mean by minimizing the movement of the object centers?

Line 325-331: Is Figure 8 consistent with the descriptions here? T2 in cell 1 is neither the last nor the first feature. What do you mean by linking the last feature of a cell to the first feature of a nearby cell? I may misunderstand something, but I need more detailed clarification of the method. In addition, could you please provide two real examples: one for merging and one for splitting? The purpose is to ensure that tobac works as you expect.

Lines 347-355: Yes, it is hard to select an appropriate distance parameter since you used the distance between features. Did you consider using the distance between segmentations? I meant the minimum distance between two segmented regions. It might eliminate the issue in Figure 9.

Lines 427-431: Why did you remap the brightness temperature dataset? According to the last paragraph, the segmentation could be conducted on the new (brightness temperature) grid. If so, plotting on the original GEOS-16 satellite grids would be more beneficial to introduce the new characteristic of tobac. In addition, in Line 430, I didn't find the top-right feature marked by the grey dot. Did you mean that in the top-left corner?

Section 4.3: Could you please provide more details about the PBC treatments? I can understand the improvements from Figures 14 and 15 but don't understand what you did to achieve them (I guess you just extended the x dimension to include the data from the next timestep in Figure 14). Is your method still valid if a feature crosses the boundary two times? In Figure 14c, the feature crosses only one boundary. If the feature is large enough, crossing another boundary on the right (or more), can tobac v1.5 get the correct result?

Lines 493-496: Please rewrite it with shorter sentences.

---

## Author Response (AR1)

**Responses to Reviewer #1**

Thank you very much for your helpful comments. They were very focused and allowed us to elaborate on details in the manuscript and of the *tobac* package itself in greater depth, and we appreciated the opportunity to improve the manuscript.

- How does the "watersheding" decide on the time when the boundary of two features meets?

Watershedding is performed on a data field independently on each timestep, and as such does not incorporate the time dimension directly. The watershedding algorithm acts by "flowing" outwards from a marker set at the feature location, with the expansion rate governed by the magnitude by which a particular grid point exceeds the segmentation threshold prescribed. Thus, where the boundaries of two or more features will meet depends both on the size of the region being segmented and the magnitudes of the data being segmented.

- Will segmentation be twisted due to the vertical structure?

 Yes, tobac can accurately segment vertically tilted or irregular structures in 3D spatial data like growing convective clouds in highly sheared environments.

- In the detection of weather phenomena like atmospheric rivers, there are thresholds in geometric shape, size, and direction, how do these thresholds be applied in tobac? In your example, magnitude thresholds are used. But for atmospheric rivers, the magnitude threshold is the very first step. Could you explain more about the process of detecting atmospheric rivers?

As the referee points out, detection of atmospheric rivers (hereafter, ARs) typically requires more than a simple magnitude-based threshold detection. *tobac* is intended to be a flexible tracker, and as such isn't specifically designed around any particular use case. However, *tobac* does provide a variety of options to allow users to tailor the feature detection to their particular phenomena of interest.

Size thresholds can be imposed for feature detection by prescribing the minimum number of contiguous grid cells that must exceed the magnitude threshold. A different value of this parameter can be set for each feature detection threshold, allowing for more flexible detection of different-sized features at different magnitudes.

Additional parameter settings and processes needed to detect ARs would likely depend on the particular case of use. One possibility is to follow a methodology similar to that used by Guan and Waliser (2019). ERA5 (they used ERA-interim) specific humidity and wind component fields can be used to calculate zonal and meridional components of Integrated Vapor Transport (IVT, see Equations 1a and 1b in Guan and Waliser, 2019). From these components, an IVT magnitude field and IVT direction field can be determined.

Guan and Waliser use a progressive application of percentile thresholds and geometry thresholds on IVT to identify coherent ARs. Percentile thresholds can straightforwardly be calculated from

the overall ERA5 IVT field and used within the feature detection step of *tobac*. Imposing the geometry thresholding after the magnitude thresholding is more complex due to the cyclical nature of wind direction, but one approach would be to perform watershed segmentation on the IVT magnitude field based on the detected features. This segmentation field can then be used to analyze the IVT zonal and meridional components, as well as the IVT direction field within the segmentation mask. Guan and Waliser prescribed a poleward (meridional) IVT component >50 kg/m/s; more than half of IVT directions in the prospective area being within 45 degrees of the mean IVT; length > 2,000 km; and a length/width ratio > 2. These former two criteria can be assessed by applying the segmentation mask to the poleward IVT field and IVT direction field; and the latter two can be assessed by just examining the dimensions of the segmentation mask itself.

If all the above criteria are met, we have detected the presence of an AR feature that can then be spatiotemporally tracked using *tobac*. Otherwise, we would either discard this potential AR feature or follow the methology of Guan and Waliser to impose higher thresholds.

**Responses to Reviewer #2**

We are very thankful for your many detailed comments on the manuscript which has improved our discussion section and specific improvements in the split and merger tool's examples and language. We also valued your clarification of details pertaining to previous tracking tools; their interest in the impetus for updating *tobac*, and your desire to explore the integrated performance and stability of all 3 key modules of *tobac* through spacetime.

"tobac v1.5: Introducing Fast 3D Tracking, Splis and Mergers, and Other Enhancements for Identifying and Analysing Meteorological Phenomena"

The manuscript is mostly well written and gives important information on the update of tobac 1.5 compared to tobac 1.2. The new features are intriguing and attractive to end users. However, my main concern is the stability of tobac 1.5, especially in 3D segmentation and tracking. In addition, the author needs to clarify how much effort the end user will need to make in order to use tobac 1.5. It seems there's quite a lot of thresholding test awaits for the users in order to use tobac 1.5, and we are not sure if the thresholds are good enough throughout the evolution of atmospheric processes. The overall paper is an useful contribution to the scientific community and requires major revision before publication.

Thank you for your overall comments and feedback. Much of your concern noted regarding stability, thresholding, and user effort depends greatly on the goal the user is looking to achieve with tobac. Most of the examples given in this publication use either radar reflectivity data or LES model output data to track convective clouds, which have different requirements for stability, effort, and thresholding than some other potential use cases (e.g., tracking of gridded precipitation fields). We elaborate on some of these details and speculate on alternative uses below.

Two key elements ultimately govern the stability of *tobac* in 3D segmentation and tracking: the distances between features in the data, and the temporal spacing of the data. As discussed in the original *tobac* publication (Heikenfeld et al., 2019), tracking performs poorly if the mean minimum distance between features is comparable to the distance a feature would be expected to propagate within one frame of time. In both the radar data and LES modeling data use cases, we were detecting and tracking convective clouds through both space and time. As such, the stability of *tobac* to track these phenomena depended on how isolated or clustered they were, and how fine the temporal spacing was. For instance, with the LES model data, we realized 15-minutely output was too coarse to reliably perform tracking on (many shallow cumulus only live for 15-30 mins; Cotton et al. 2011), so we increased the resolution to 5 minutes, which was able to address this task more robustly. Segmentation followed a similar line of thinking – too coarse of a temporal resolution will not allow one to accurately capture the evolution of a cloud and its boundaries (defined by the total condensate mixing ratio).

The thresholding tests and requisite effort needed to use *tobac* are similarly dependent on the use case. For example, for the identification, segmentation, and tracking of clouds in LES model data, we used four thresholds of 1.e-5, 1.e-4, 1.e-3, and 1.e-2 g/kg total condensate mixing ratio. These thresholds were selected based on previously published values and thresholds used for

cloud liquid water content (e.g., Cotton et al. 2011; Sheffield and van den Heever 2015) and are evenly spaced in logarithmic space. However, we did not need to perform any additional testing of these thresholds since previous studies had established their physical significance and they produced the expected *tobac* output. In many cases where the user has familiarity with the data, its dynamic range, and the phenomena they are looking to identify, we do not expect much testing of thresholds to be needed. However, in some other cases where a user is looking to isolate something specific in the data (e.g., identification of the certain presence of hail in radar reflectivity data), we agree that more experimentation of appropriate thresholds would be needed.

1. Introduction section, the authors keep referring to tracking object as "cloud", as Tobac is largely applied to radar cell tracking, please make sure to clarify radar does not see cloud (most X-, C-, S-bands radars) but only observe rain droplets and hydrometeors that are much larger than cloud droplets.

Although radar data are one of the many datasets *tobac* can be used with, it can also be used with essentially any gridded data set, such as numerical model output (Figures 2, 5, 14, 15) and satellite observations (Figure 13). As such, we have chosen to retain the "cloud" as an appropriate overarching terminology to describe our targets of interest since we believe this most accurately describes the range of *tobac*'s capabilities.

2. Line 70, regarding the introduction of TITAN, the major disadvantage is its centroid method based on reflectivity and even the modern version of TITAN allowing multiple thresholds perform not well for tracking storm cells from initiation to the very end.

Thank you for pointing this out. We have now clarified this in the text in lines 73-74:

"…limitations at the time and is a centroid-based method which sometimes has difficulty tracking storm systems for their full lifetimes."

3. Line 95, the WDSS II system used SCIT as its default tracking toolbox and WDSS II can be open-source base on request. The main flaw of SCIT is similar to TITAN as they are both centroid based methods.

Thanks for this comment. We have clarified the discussion of WDSS II in the manuscript at lines (95-100):

"The Warning Decision Support System–Integrated Information (WDSS-II) data synthesis platform (Lakshmanan et al., 2007) includes multiple tracking packages, including the Storm Cell Identification and Tracking algorithm (Johnson et al. 1998), a multi-scale cell tracking algorithm, and cross-sensor fusion capability (Lakshmanan et al., 2009; Lakshmanan and Smith, 2009). WDSS-II has been widely used for real-time applications in the US National Weather Service, but is subject to licensing restrictions for that purpose, although its source code is apparently available upon request."

4. Line 110-115, please elaborate on the reasoning behind this Tobac upgrade here, is it the tobac v1.2 cannot handle high resolution? Why the new missions like AOS and INCUS will require this upgrade? I see you listed a lot of new features in the upgraded tobac, please elaborate how these new features can contribute to cell tracking.

Version 1.2 of *tobac* lacks processing of the vertical dimension, periodic boundary conditions, split and merger processing, and a spectral filtering tool which lend a great deal of additional use to high-resolution modeling (e.g. LES) and observational datasets. *tobac* v1.2 is also considerably less efficient at processing data than v1.5, as shown in Figure 12. New missions such as AOS and INCUS require these upgrades due to the vast quantities of high-resolution 3D data the missions and algorithm development will produce. We have updated the language here, in lines (116-122):

"Despite the utility, modularity, and flexibility of *tobac* v1.2, the increasing resolution and spatial extent of models and identification of new use cases (such as in LES modelling made it clear that the code base needed to be enhanced from both a scientific and procedural point of view. The advent of new spaceborne missions with high-resolution observations, such as the National Aeronautics and Space Administration's Atmospheric Observing System (AOS) and Investigation of Convective Updrafts (INCUS) programs and the European Space Agency's EarthCARE program, will involve the collection of vast quantities of 3D data that require processing of the vertical dimension with great efficiency that *tobac* v1.2 cannot do."

5. Figure 1, is tobac v1.2 and tobac 1.5 both centroid based methods? If so, does that means the anvil part of the cell is missing as shown in figure 1? In addition, what kind of QC has been applied to the reflectivity dataset? SNR? Attenuation correction? And maybe self-consistency calibration?

*tobac* v1.2 and v1.5 are indeed both centroid-based methods for feature identification and tracking; segmentation uses the watershedding method, growing out from the centroid as described above. Figure 1 is intended to be a simple demonstration of *tobac*'s process, and as such, detailed QC was not performed before gridding the data. This figure shows reflectivity at a slice 2 km above ground level after gridding with PyART, and as such wouldn't include anvils. We've updated the caption to clarify this:

"Figure 1: Demonstration of *tobac* feature detection and segmentation of gridded NEXRAD radar reflectivity data at 2 km above ground level from the Cheyenne, WY radar (located just to the NW of the domain shown here) on 25 May 2017 during the C$^3$LOUD-Ex field campaign (van den Heever et al., 2021)…"

6. Line 144-157, it seems tobac is, in general, a centroid based method as the latest TITAN. So any feature selection such the size of object, the reflectivity thresholds, and more do require " a great deal of human input and attention" as mentioned in the introduction when the authors are reviewing other tracking methods.

As the reviewer points out, *tobac* does represent features as points, and some human attention is required for the use of *tobac* to ensure that module parameters and thresholds have been appropriately configured to perform any of feature detection, segmentation, or tracking.

This said, many of the tools discussed in the introduction do indeed require "a great deal of human input and attention" – especially when taking into account *tobac*'s modularity and variable-agnosticity. Dawe and Austin (2012) discuss a number of earlier tracking tools with such limitations, and some that we explicitly mention are also more labor-intensive. TITAN, for example, requires specific variables in order to be fully utilized, and the method of Heus and Seifert (2013) requires users to define all of thermals, cloud envelopes, and precipitation shafts from their source data.

In summary, as with many data analysis tools, *tobac* does need some user-configured options, but it can be used on a much greater variety of datasets with less a-priori configuration of the data than most of the tools discussed in the introduction.

7. Line 163-174, using watershed-based method for segmentation can be quite useful here, but the over or under segmentation happens often. Please share multiple (at least 5-time steps) before and after segmentation figures for the case shown in Figure 1. I'm curious to see how tobac segmentation performs in terms of stability here.

While we understand the reviewer's interest in seeing the performance of *tobac* segmentation in greater detail, the purpose of Figure 1 is simply to demonstrate the performance of the feature detection and segmentation modules of *tobac* as implemented in *tobac* v1.2.

Instead of modifying Figure 1 in the manuscript, we prepared an example of *tobac* feature detection, segmentation, and tracking on an MRMS hourly composite reflectivity dataset in Figure R1 below.

[Figure]

Figure R1: *tobac* feature detection, segmentation, and tracking on hourly MRMS (Multi-Radar Multi-Sensor) composite reflectivity data from 1400Z to 1800Z on 31 March 2023. Panels (a) through (e) depict the composite reflectivity field; panels (f) through (j) depict the reflectivity field overlaid with *tobac*-detected features (colored dots), the segmentation masks associated with those features (white outlines), and the tracked cell paths associated with the features (black lines). Feature detection was thresholded on 40 (yellow dot), 50 (red dot), and 60 (magenta dot) dBZ. Segmentation was thresholded on 20 dBZ; and tracking restricted features in adjacent temporal frames to a maximum estimated velocity of 15 m/s (parameter 'vmax' = 15; search range of 54 km for hourly data) for being linked together into a *tobac*-identified cell. NOAA Multi-Radar/Multi-Sensor System (MRMS) was accessed on 10 February 2024 from https://registry.opendata.aws/noaa-mrms-pds.

While this is an illustration using tobac v1.5 and not v1.2, it does not demonstrate any of the new features present in v1.5 and should suitably illustrate the stability of tobac segmentation.

The potential analyses and associated QC considerations (e.g., stability of segmentation over time) resulting from the combination of results from discrete modules of *tobac* has been demonstrated more extensively in recent studies, such as that by Oue et al. (2022, AMT).

8. Line 203-220, base on the example and text here, tobac is a user defined centroid based method. It is counter productive to use hard thresholds while using watershed-based method. This is just an opinion/comment and does not require authors to reply.

One of the intentional design decisions made with *tobac* was to make it extensible. Future versions of *tobac* will include feature detection and segmentation methods other than absolute thresholds, but the procedures and code will remain in the same framework.

9. Figure 3, tobac 1.5 has a nice touch with multi-layer clouds. It is clear this can be used in model outputs. Any idea what kind of observational data we can use to test the performance of tobac 1.5 on multi-layer clouds? I'm guessing not polar coordinate satellites as the temporal resolution is low, but geostationary satellites such as GOES-series are all passive sensors and cannot provide 3D structure features for tracking. It is also hard for most radars as shorter wavelengths suffer from attenuation but longer wavelengths are not sensitive enough to observe cirrus clouds.

Ground-based radar data may offer some possibilities in this regard, as some such radar platforms do have transmitters which operate at bands that can see cloud particles.

10. Figure 5, how sensitive is the 3D segmentation with boxing method is to user picked size? Please include seed size from 2,4,6, and 8 in the reply. I'm curious how stable this boxing method is here. Do the authors believe that changing the size of the box will greatly impact the quality of tracking here? Especially during different stages of cell involvement.

We chose to test segmentation using seed sizes of 1, 3, 5, 7, and 9 – this covers the full range of box sizes the reviewer is interested in, and is a more physically reasonable approach given the grid structure and package behavior. *tobac* identifies represents features as individual points, which can have coordinates in decimal space that do not correspond precisely to a grid point. The segmentation module places the feature at the nearest integer grid point to perform watershedding. Thus, a box size of 1 corresponds to simply using the feature gridpoint itself as the marker, where box sizes of 3, 5, 7, and 9 all progressively expand the size of the marker by one grid cell. Odd numbers are required so as to not bias the box to one side or the other.

We have included a plot of these results as follows:

[Figure]

Figure R2: Demonstration of differences in performance of 3D watershed segmentation with different seeding box sizes. Each of panels (a) through (e) depict odd-numbered increments of seed box sizes ranging from 1 through 9. The detected watershed volumes for this feature are identical regardless of the seed box size used.

For the use case demonstrated in Figure R2, our segmentation volumes are identical. In general, however, the stability of the method with different box sizes depends greatly on the number and proximity of features and the size and contiguity of the field being segmented. In a field of isolated shallow cumuli, altering the box size would likely have little impact, whereas segmenting different convective cells in a multicellular storm system would likely depend a lot on the box sizes – too large of a box could artificially designate one cell as being part of an adjacent cell. We have currently set a box size of 5 (5 points in each of the x, y, and z dimensions) as the default based on experience.

Altering the size of the box seeding used in segmentation would not have any effect on tracking performance. Tracking is currently performed independently from segmentation on previously detected features and does not require segmentation to have been performed.

11. Figure 6, please add contours of the cell segmentation to all panels. In addition, the authors mention the user need to pick Z threshold here, demonstrated in 30 dBZ. What will happen if the user wishes to include all the stratiform portion of the clouds and set Z limit to – 10 dBZ? Will tobac still be able to segment? Please also include the 3D view of panel e-h, I'm curious how the regeneration of cells as it propagates will impact 3D segmentation here, or if there is any?

After examining the figure plotted using the additional contours suggested by the reviewer, we have decided to keep Figure 6 as it was, since adding the cell segmentation contours results in too cluttered of a figure. This figure illustrates the performance of the tracking module, which operates independently from segmentation, and in fact does not require segmentation to be completed. The 3D updates to the tracking module more closely align with (and depend on) those made to the feature detection module and are wholly independent of any 3D improvements to or options selected (e.g., threshold magnitude) for segmentation.

For the case suggested by the reviewer, if the user wishes to include all of the stratiform portion of the clouds as they are tracked through space-time, this is something tobac should indeed be successful at doing. We would suggest retaining a higher reflectivity threshold for feature detection (e.g. 30 dBZ) while using a lower threshold such as 0 dBZ or -10 dBZ to capture the stratiform region of the cloud via segmentation. As long as the stratiform portion of the cloud is connected in gridpoint space to the location where the parent feature was detected, segmentation will appropriately capture the stratiform regions. Challenges may arise in cases with large-scale, organized convection containing many discrete convective cells (e.g., Mesoscale Convective Complexes) or if the threshold for feature detection is set too low (as this may result in the identification of non-convective phenomena and/or designating too large of a region as being associated with a feature.

12. Line 301, "Hu et al." typo, missing the year.

We have corrected this, thank you.

13. Figure 9. Please add 4 time steps of 2D segmentation and tracking (with splits and mergers labelled) to this case. Also, what will happen if one uses 0 dBZ here as threshold.

If 0 dBZ were used as a threshold, *tobac* detection and tracking would be much more sensitive to the presence of features relative to the 30 dBZ threshold, which would probably limit the meaningful analysis that can be performed in a QLCS case such as that shown in the original Figure 9. For example, mature squall lines generally consist of a leading convective line with heavy precipitation, followed by a trailing stratiform region with lighter precipitation. Barring a contiguity gap between these two regions in radar data, *tobac* would probably reduce the QLCS

structure to a single point feature, making it difficult to study different parts of the squall line or their evolution.

In response to the requested changes to Figure 9 and a similar question from Reviewer 3, we have generated a new Figure 9 and a new merge demonstration figure to be included in an appendix (Figure A1). Hopefully these are sufficient to demonstrate a split and merge in practice on actual data.

[Figure]

Figure 9: MRMS (Multi-Radar Multi-Sensor) hourly maximum composite reflectivity data depicting a splitting convective system near the Texas-Oklahoma border on 31 March 2018. The left column depicts the MRMS data; right column depicts MRMS data overlaid with detected features (colored dots), segmentation masks (white outlines), tracked cells (black lines), and cell and parent track labels (black and white numbers, respectively) Cell track 1038 (parent track 8) does not meet the split or merge criteria with cell tracks 1037 or 1115 (parent track 7),

whereas Cells 1037 and 1115 are determined to have split from one another. Feature detection was thresholded on 40 (yellow dot), 50 (red dot), and 60 (magenta dot) dBZ. Segmentation was thresholded on 20 dBZ; and tracking restricted features in adjacent temporal frames to a maximum estimated velocity of 15 m/s.

[Figure]

Figure A1: A visualisation of 4 frames (a-d) of a *tobac*-tracked and detected merger from the KHTX NEXRAD radar site at 19:19:03Z to 19:32:15Z on 30 April 2016. The cell number is given in red, the parent track ID is in blue, and the feature locations at the present timestep is marked with a dot in each panel. Initially, Cells 528 and 555 are both present (panel (a)). However, over the course of their evolution, the tracks can be seen to merge together (Panels (b-c)) as Parent Track 155, with Cell 528 no longer existing in Panel (d).

We also revised the body text to provide a description of the new Figure 9 (lines 356-357):

"In Figure 9, we demonstrate the procedure in use on real MRMS hourly composite reflectivity data and have detected a split occurring during the evolution of a convective cell."

> 14. Sec 4.1. What is the 3D tracking efficiency here using tobac 1.5? Let's say using MRMS gridded 3D Z field for 1 day using Z threshold of 15 dBZ.

Section 4.1 is primarily intended to demonstrate the improvement of *tobac*'s efficiency in v1.5 relative to v1.2, so we have provided specific examples in the text and Figure 12 to explicitly draw this contrast between elements of *tobac* that exist in both versions. Since 3D tracking was introduced in *tobac* v1.5 along with the efficiency improvements, there is no baseline efficiency in *tobac* v1.2 to draw a contrast to. The performance of 3D feature detection and tracking on such a case will depend not only on the user's system hardware, but also the number of features in the dataset being examined. As a result, performing this analysis on two different MRMS datasets with identical dimensional structure (i.e., time, x, y, and z points) would have radically different efficiencies. From experience, this sort of analysis on a 5-minute dataset can be completed in a few hours on a reasonable server. The bottleneck in feature detection is typically the user's available memory.

**Responses to Reviewer #3**

We thank the reviewer for their detailed and helpful comments and suggestions for the manuscript. All helped to clarify our presentation of the material, and we were especially appreciative of the comments pertaining to the split and merge tool and the PBC procedure. We understand that the initial discussion pertaining to these areas specifically was complex and challenging to follow in some places, so we welcomed the opportunity to expand upon our work.

Line 1-3: Could you please use a better title?

We appreciate the reviewer's interest in seeing an improved title. Under GMD submission guidelines, we are required to put the name and version number of the software tool before providing any other details in the title, so we cannot change this part of it. While we understand that our title is wordy, we believe this was the most succinct yet comprehensive way to address the improvements to *tobac* and their purpose covered in the manuscript.

Line 32: Delete "diffusive, advective"? Do you think diffusion and advection are neither dynamic nor thermodynamic processes?

Thank you for your comment, we have removed this from the text.

Lines 281-282: I don't understand the logic of this sentence. If there are just a few examples of splits and mergers in the atmosphere, why is there a clear need for splits and mergers processing within tobac?

We realize this statement was worded in a confusing way. It is not that there are few examples of splits and mergers in the atmosphere, it is that we only provide a few examples in the text. We have clarified this passage at lines (289-291) to read:

"These examples are just a few of the many processes involving splits and mergers in the atmosphere, and thus there is a clear need for splits and merger processing within *tobac*.[…]"

Line 292: What do you mean by minimizing the movement of the object centers?

This sentence was intended to summarize part of the approach used by Dixon and Weiner (1993) for tracking in TITAN. For each time step, Dixon and Weiner predict the possible motion of all detected feature centroids, resulting in many potential tracks. The first step in their determination of the actual path is the consideration that "the correct set will include paths that are shorter rather than longer". As such, we initially summarized this process by referring to it as "minimizing the movement of the object centers.", but have rephrased this to say "selecting the shortest, and thus most likely, path.", now on line (301).

Line 325-331: Is Figure 8 consistent with the descriptions here? T2 in cell 1 is neither the last nor the first feature. What do you mean by linking the last feature of a cell to the first feature of a nearby cell? I may misunderstand something, but I need more detailed clarification of the

method. In addition, could you please provide two real examples: one for merging and one for splitting? The purpose is to ensure that tobac works as you expect.

Thank you for identifying this inconsistency between text and figure. Our initial text did not properly describe some of the details of the procedure, and we apologize for the ensuing confusion. For split/merger determination, we allow a user-set window (the default is 5 timesteps) with which to match the last feature of one cell to the start of a nearby cell. The end of one cell can occur after the start of the nearby cell, and as long as those fall within the 5 timesteps (or whatever the user decides) and the distance thresholds, they can match. Thus, in Figure 8, where the T1 of Cell 2 has a line drawn into the T2 of Cell 1, it would technically be 'more' correct to draw the line between T0 of Cell 2 to Cell 1. We have depicted this below in the revised Figure 8, which has also been updated in the manuscript.

[Figure]

Figure 8: An illustration of merging cells (Cells 1 and 2) and a standalone cell (Cell 3) as perceived by tobac. All three cells are comprised of features in radar data which exceeded a 15 dBZ threshold. Merging criteria (size and proximity) for the "tail" of Cell 1 and "tip" of Cell 2 are met at time t2, thus, these cells are judged to have merged over their lifetimes.

We have also changed the text to give a more accurate description of the process:

"works by linking the last feature of a cell to the first feature of a nearby cell. We take the last feature of all cells, and then find the distance between each last feature and each initial feature within a user-set number of timesteps (the default value is 5) for all timesteps. This distance is

the weight of the branches in the MEDST. Before further processing these paired points (i.e., the location of the last feature in a cell and the additional feature in another cell) […]" (335-338).

Additionally, we have produced a new Figure 9 in the manuscript as well as a Supplemental figure, Figure S1, in response to your comment and a similar comment from Reviewer 2 interested in seeing more timesteps of the split and merge processes. We hope these examples are sufficient to demonstrate that the split/merge tool works as intended.

Lines 347-355: Yes, it is hard to select an appropriate distance parameter since you used the distance between features. Did you consider using the distance between segmentations? I meant the minimum distance between two segmented regions. It might eliminate the issue in Figure 9.

Thank you for this great suggestion. We have not yet considered using the distance between segmentation regions for split and merge criteria in the present implementation within *tobac*. This is now being explored for future versions.

Lines 427-431: Why did you remap the brightness temperature dataset? According to the last paragraph, the segmentation could be conducted on the new (brightness temperature) grid. If so, plotting on the original GEOS-16 satellite grids would be more beneficial to introduce the new characteristic of tobac. In addition, in Line 430, I didn't find the top-right feature marked by the grey dot. Did you mean that in the top-left corner?

We disagree with the reviewer that performing segmentation on the brightness temperature grid would have better demonstrated this new aspect of *tobac*. As feature detection was initially performed on the radar data grid, we feel that remapping the brightness temperature dataset so we could perform segmentation on the same grid that feature detection was performed on is more appropriate for analysis purposes. Further, the GOES-16 grid is more spatially coarse than the NEXRAD data, so remapping the detected features and/or radar data to this grid would lose some numerical precision.

Additionally, we appreciate your identification that Figure 13 does not match the associated text in the manuscript, and we have adjusted the text to remove any mention of the absent feature:

"Ultimately, the segmentation outlines shown in Fig. 13b depict the anvils corresponding to each marked radar reflectivity feature." (440-441)

Section 4.3: Could you please provide more details about the PBC treatments? I can understand the improvements from Figures 14 and 15 but don't understand what you did to achieve them (I guess you just extended the x dimension to include the data from the next timestep in Figure 14). Is your method still valid if a feature crosses the boundary two times? In Figure 14c, the feature crosses only one boundary. If the feature is large enough, crossing another boundary on the right (or more), can tobac v1.5 get the correct result?

Thank you for your comment. This was actually a complex undertaking which required different approaches for each of feature detection, segmentation, and tracking. As such, we avoid spending too much time focusing on this treatment in the manuscript due to the number of other

improvements in v1.5 which needed to be discussed. We have provided a more detailed explanation of our approach here in our response to you and have also prepared a short appendix, but are reluctant to add extensive additional text to the primary manuscript given the length of the existing text.

Once labeling of data fields is performed during the feature detection treatment, the PBC routine "looks" across one or both model boundaries to see if there are any labeled regions which are contiguous, but artificially separated by the model boundaries. Overwriting of labels in eligible data regions is performed continuously until all contiguous data regions have their own discrete label.

Tracking handles PBCs by implementing a custom distance function that removes the "wrap-around" distances found across these boundaries.

The PBC treatment for segmentation is more complex. We provide an illustration of two particular use cases the treatment has been designed to address in Figure B1 below:

[Figure]

Figure B1: A depiction of two elements the PBC segmentation treatment. In Panel (a), we have two sets of regions (the red and blue shapes, and the magenta and white shapes) which are artificially separated by a model boundary. The white shape in Panel (a) is "eligible but unseeded", meaning it exceeds the segmentation magnitude threshold but did not have a marker placed within it to conduct watershedding. Since the magenta shape is in contact with this shape across the model boundary, we first seed all adjacent boundary points (as depicted in Panel (b) ) and then use these to watershed the relevant "eligible but unseeded points" as shown in Panel (c). The red and blue shapes shown in Panel (a) have both been watershed by different feature markers, but are artificially separated by the model boundary. This necessitates the selection of these two shapes into their own contiguous domain (the dashed orange "Buddy Box" depicted in Panels (c)-(f) ) so that watershedding can be performed again to obtain the correct segmentation boundaries. After transforming these grid points and their included data into the "Buddy Box" domain as shown in Panel (d), we place our feature markers (the red and blue boxes with black outlines) in the domain and perform watershedding again, as shown in Panel (e). Subsequent to the "Buddy Box" watershedding, the correct segmentation regions are transformed back into the original domain, as shown in Panel (f). Boxes with bright colors and black outlines depict the feature markers used for watershedding; the paler corresponding colors denote the regions segmented by these markers through watershedding; the white boxes denote "eligible but unseeded" regions

(i.e., above the segmentation magnitude threshold but not marked by a feature), and the gray boxes denote regions that are beneath the magnitude threshold and are not eligible to undergo watershedding.

The simplest case occurs when a segmentation field on one side of a boundary has been marked and watershed, but the same contiguous field on the other side of that boundary has not (the magenta and white shapes in Figure B1(a)). Here, we simply assign the same label to the unlabeled region as that which has already been watershed (Figure B1(b)-(c)).

For cases when two or more labeled segmentation regions meet at a model boundary (the blue and red shapes in Figure B1(a)), a new sub-domain (the dashed orange outline, which we refer to as a "Buddy Box" in this figure and our source code) is constructed of only the grid cells corresponding to these segmentation regions, but with all artificial boundaries removed so the fields are continuous. Then, markers are placed again (Panel (d) ) and watershed segmentation is performed on this new sub-domain (Panel (e) ). The segmentation boundaries ensuing from this procedure are then adapted back to the original segmentation field before PBC treatment, as shown in Panel (f).

Lines 493-496: Please rewrite it with shorter sentences.

Thank you for this comment, we have broken up this passage into shorter sentences as shown below:

"One key element planned for the next major release includes integration with the TiNT is not TITAN (TiNT; Raut et al., 2021) tracking package. We are also seeking to transition away from tobac's current memory-intensive data structures to data structures that allow for out-of-memory computation instead (e.g., Dask from Rocklin, 2015; xarray from Hoyer and Hamman, 2017)." (503-506)

The manuscript is a pleasant read and I congratulate the authors on the substantial improvements made to a promising, versatile tool for meteorological feature tracking. I just have a few minor comments as a community member.

Thank you for your kind words and thoughtful comments. They have improved the manuscript, particularly regarding our consideration of Atmospheric Rivers and discussion in the introduction.

1. The manuscript did not mention whether the detection/identification of features (that is, the step before segmentation and tracking) can be based on multiple input fields (not just multiple thresholds for the same field) or can only take one input field in that step. For reference, in the case of atmospheric rivers (ARs), there are algorithms that rely on two or more of the following fields to detect the features: integrated water vapor (IWV), IWV transport, wind, precipitation, etc.

Thank you for your thoughtful comment. We agree with the reviewer that this would be extremely useful. Unfortunately, at this point in time, this version of *tobac* cannot be used on multiple input fields without multiple feature detection steps. However, implementing multivariate tracking while preserving our ability to remain variable-agnostic is already a key feature of our upcoming development plans for *tobac*, and initial results in the implementation of these capabilities are promising.

2. Some of the improvements introduced in the manuscript have also been introduced in AR tracking algorithms, such as handling of separations and mergers, periodic boundary conditions, and grid agnosticity (Guan & Waliser, 2015, 2019).

Thank you for mentioning these publications. We were remiss in not citing these papers, and now include a description of their findings as well as the appropriate citations as shown below:

"Guan and Waliser (2015, 2019) have developed a tool called Tracking Atmospheric Rivers Globally as Elongated Targets (TARGET), which is designed for the detection and tracking of atmospheric rivers (ARs). TARGET includes techniques such as split and merger processing; periodic boundary condition treatments; and grid agnosticity; but can only be applied as presently designed to ARs." (105-108)

3. Given the number of studies dedicated to AR detection in recent years, including those contributing to the Atmospheric River Tracking Method Intercomparison Project (ARTMIP; Shields et al., 2018), some general discussion of AR detection might be helpful to provide more context and motivation for the development of *tobac*.

While we agree that AR detection and tracking are very important, we were reluctant to spend too much time expanding upon this particular use case due to the multitude of other atmospheric phenomena that can be detected with tobac, and the variety of improvements that needed to be

discussed in the paper. Nonetheless, we have added a mention of ARTMIP and short discussion of the innovative techniques in TARGET:

"In recent years, there has also been a greater research focus on Atmospheric Rivers (ARs), including many existing within the Atmospheric River Tracking Method Intercomparison Project (ARTMIP; Shields et al. 2018). Guan and Waliser (2015, 2019) have developed a tool called Tracking Atmospheric Rivers Globally as Elongated Targets (TARGET), which is designed for the detection and tracking of atmospheric rivers (ARs). TARGET includes techniques such as split and merger processing; periodic boundary condition treatments; and grid agnosticity; but can only be applied as presently designed to ARs." (103-108)

References

Cotton, W. R., Bryan, G. H., and van den Heever, S. C.: Storm and Cloud Dynamics, Academic press, 2011.

Oue, M., Saleeby, S. M., Marinescu, P. J., Kollias, P., and van den Heever, S. C.: Optimizing radar scan strategies for tracking isolated deep convection using observing system simulation experiments, Atmos. Meas. Tech., 15, 4931–4950, https://doi.org/10.5194/amt-15-4931-2022, 2022.

Sheffield, A. M., Saleeby, S. M., and van den Heever, S. C.: Aerosol-induced mechanisms for cumulus congestus growth. J. Geophys. Res., 120, 8941–8952, 2015.